# Constitutive activation of the PI3K-Akt-mTORC1 pathway sustains the m.3243 A > G mtDNA mutation

Chih-Yao Chung [1], Kritarth Singh[1], Vassilios N. Kotiadis[1], Gabriel E. Valdebenito [1], Jee Hwan Ahn [1], Emilie Topley[1], Joycelyn Tan [1], William D. Andrews[1], Benoit Bilanges[2], Robert D. S. Pitceathly [3], Gyorgy Szabadkai [1,4,5], Mariia Yuneva [5] & Michael R. Duchen [1✉]

Mutations of the mitochondrial genome (mtDNA) cause a range of profoundly debilitating clinical conditions for which treatment options are very limited. Most mtDNA diseases show heteroplasmy – tissues express both wild-type and mutant mtDNA. While the level of heteroplasmy broadly correlates with disease severity, the relationships between specific mtDNA mutations, heteroplasmy, disease phenotype and severity are poorly understood. We have carried out extensive bioenergetic, metabolomic and RNAseq studies on heteroplasmic patient-derived cells carrying the most prevalent disease related mtDNA mutation, the m.3243 A > G. These studies reveal that the mutation promotes changes in metabolites which are associated with the upregulation of the PI3K-Akt-mTORC1 axis in patient-derived cells and tissues. Remarkably, pharmacological inhibition of PI3K, Akt, or mTORC1 reduced mtDNA mutant load and partially rescued cellular bioenergetic function. The PI3K-Akt-mTORC1 axis thus represents a potential therapeutic target that may benefit people suffering from the consequences of the m.3243 A > G mutation.

[1] Department of Cell and Developmental Biology and Consortium for Mitochondrial Research, UCL, Gower Street, London WC1E 6BT, UK. [2] UCL Cancer Institute, 72 Huntley St, London WC1E 6DD, UK. [3] Department of Neuromuscular Diseases, UCL Queen Square Institute of Neurology and The National Hospital for Neurology and Neurosurgery, Queen Square, London WC1N 3BG, UK. [4] Department of Biomedical Sciences, University of Padua, via G. Colombo 3, 35100 Padua, Italy. [5] The Francis Crick Institute, 1 Midland Rd, London NW1 1AT, UK. ✉email: m.duchen@ucl.ac.uk

Mitochondria control cellular bioenergetic homoeostasis and serve as a hub for cell metabolism and cell signalling pathways. Human mitochondria contain a circular plasmid-like DNA (mtDNA) which encodes 13 proteins that act as subunits of the electron transport chain (ETC) and 24 RNAs essential for mitochondrial protein synthesis. Mutations of mtDNA affect around 1 in 5000 of the population[1] and cause a range of diseases for which no effective treatment is available[2,3]. Over half of the known pathogenic mutations are found within tRNA genes, in which the m.3243 A > G mutation, a tRNA$^{Leu}$ point mutation accounts for about 40% of adult patients with primary mitochondrial diseases[1,4]. Clinically, signs and symptoms in patients with mtDNA mutations are highly heterogeneous[2,5]. The tissues primarily affected vary depending on the specific mtDNA mutations, and our understanding of the relationships between mtDNA mutations, disease phenotype and severity is very limited[2,3]. The majority of diseases caused by mutations of mtDNA are heteroplasmic—tissues express both normal and mutation-carrying mtDNA. This is a confounding complication, as disease expression may differ radically between patients with the same mutation but with a different mutant load. While disease severity broadly correlates positively with the relative burden of mutant mtDNA, we know remarkably little about the determinants of the mutant mtDNA burden[3,6–8]. We, therefore, asked how cell signalling pathways influence these pathways in the disease model, as adaptive - or maladaptive—responses to impaired oxidative phosphorylation (OxPhos) and changes in intermediary metabolism, and whether these pathways play a role as determinants of mutant load and disease severity.

In this study, we have characterised the metabolic phenotype of patient-derived cells bearing the m.3243 A > G (tRNA$^{Leu}$) mutation. This is the most common heteroplasmic mtDNA mutation[1,4], and, clinically, is expressed variably but may include diabetes, sensorineural deafness, myopathy, encephalopathy, lactic acidosis and stroke-like episodes (MELAS). We have found that the basal expression and activity of the PI3K-Akt-mTORC1 pathway were increased in the mutant cells, and were strongly associated with redox imbalance, oxidative stress, and glucose dependence. Phosphorylation of Akt and ribosomal protein S6 (a downstream target of mTORC1) were increased in muscle biopsies from other patients with the mutation, confirming that this signalling pathway is constitutively activated in patient tissues. Remarkably, inhibition of PI3K, Akt, or mTORC1 all substantially reduced mutant load and improved mitochondrial bioenergetic function. These findings thus reveal that in response to the m.3243 A > G mutation, metabolism is rewired, and activity of the PI3K-Akt-mTORC1 axis is increased, presumably as an adaptive response to metabolic changes driven by the mutation. The finding that inhibition of the pathway reduces mutant load and improves mitochondrial bioenergetic function suggests that activation of this signalling pathway is, in fact, maladaptive, that activation of the pathway sustains disease progression and that pharmacological intervention in this signalling pathway represents a potential therapeutic strategy in patients with these dreadful diseases.

## Results

### The m.3243 A > G mutation causes mitochondrial dysfunction, increasing oxidative stress

To explore the metabolic and cell signalling impact of the m.3243 A > G mutation, we used six cell lines: fibroblasts derived from two patients carrying the mutation, two controls matched for age and gender, an A549 cybrid cell line carrying the mutation and its wild-type (WT) counterpart. PCR-RFLP[9] was used to ensure the presence of the m.3243 A > G mutation and ARMS-qPCR[10] was used to quantify the m.3243

A > G mutation load in these cell models. One patient-derived line (henceforth referred to as patient 1) showed a mutant load of 86.2 ± 2.3%; a second line (referred to as patient 2) showed a mutant load of 30.3 ± 3.5% and the mutant load was 79.0 ± 0.3% in the cybrid cells (Fig. 1a; for more details about the patients please see Methods).

To characterise the metabolic phenotype of the mutant cells, respiratory rate was measured using the Seahorse XFe96 extracellular flux analyser. These measurements showed a profound decrease in resting respiratory rate, ATP-dependent respiration and maximal respiratory capacity, and an increase in extracellular acidification rate (ECAR) in both patient lines (Fig. 1b–d). Consistent with these findings, immunoblotting of respiratory chain supercomplexes using blue native gel electrophoresis (BNGE), which identifies non-denatured macromolecular assemblies, revealed disrupted expression of respiratory chain proteins in the mutant cells (Fig. 1e, f). Notably, there was a very large decrease in the assembly of supercomplexes $I_2 + III_2 + IV_n$, $III + IV_1$ and $III_2/IV_2$ and in complex $IV_1$. Surprisingly, levels of complex II, which is entirely encoded by nuclear genes, were also consistently and significantly reduced in all mutant cell lines. Mitochondrial membrane potential ($\Delta\psi_m$) was reduced in both patient-derived fibroblast lines (Fig. 1g, h). The decrease in $\Delta\psi_m$ was greater in cells from patient 1 than from patient 2, consistent with the greater mutant load in patient 1. Mitochondrial dysfunction may alter rates of free radical generation by the respiratory chain. The rate of increase of dihydroethidium (DHE) fluorescence, reflecting the intracellular rate of production of reactive oxygen species (ROS), was significantly increased both in patient fibroblasts compared to matched controls (Fig. 1i). MitoSOX, a mitochondria-targeted form of DHE, revealed a significant increase in the rate of ROS generation in the mitochondria in the mutant cells (Fig. 1i), suggesting that the increased rate of ROS generation is likely the consequence of an impaired respiratory chain. These metabolic features were recapitulated in the A549 cybrid cells (Supplementary Fig. 1), including OCR (since OCR is affected by cellular mitochondrial content, its value was normalised to their increased mtDNA copy number, Supplementary Fig. 1b), ECAR, and $\Delta\psi_m$. Together, the m.3243 A > G mutation results in mitochondrial dysfunction and elevated ROS generation.

### Increased glucose uptake and dependence, resulting in redox imbalance, are features of cells bearing the m.3243 A > G mutation

We wondered whether glucose or glutamine metabolism is altered in the mutant cells, supporting cellular bioenergetic homoeostasis and compensating for the mitochondrial dysfunction. Using time-lapse live-cell imaging, we quantified cell growth rates by fitting the curves of cell confluence with an exponential cell growth model (Supplementary Fig. 2a) under a variety of nutrient conditions. Growth rates were significantly reduced in all mutant fibroblasts compared to their control counterparts (Supplementary Fig. 2b). Moreover, the cell proliferation rate was significantly reduced in all mutant cells in 1 and 5 mM glucose media (Fig. 2a, b) but not in low concentrations of glutamine, suggesting that the patient-derived cells are significantly more dependent on glucose metabolism. Consistently, the rate of uptake of the fluorescent glucose analogue, 2-NBDG, was significantly increased in patient fibroblasts compared to controls (Fig. 2c).

The cytosolic NADH:NAD$^+$ ratio is a function of glycolytic flux (which consumes NAD$^+$, generating NADH), lactate production through lactate dehydrogenase (LDH) which produces lactate and regenerates NAD$^+$ from NADH and the activity of malate/aspartate and the glycerol phosphate shuttles that

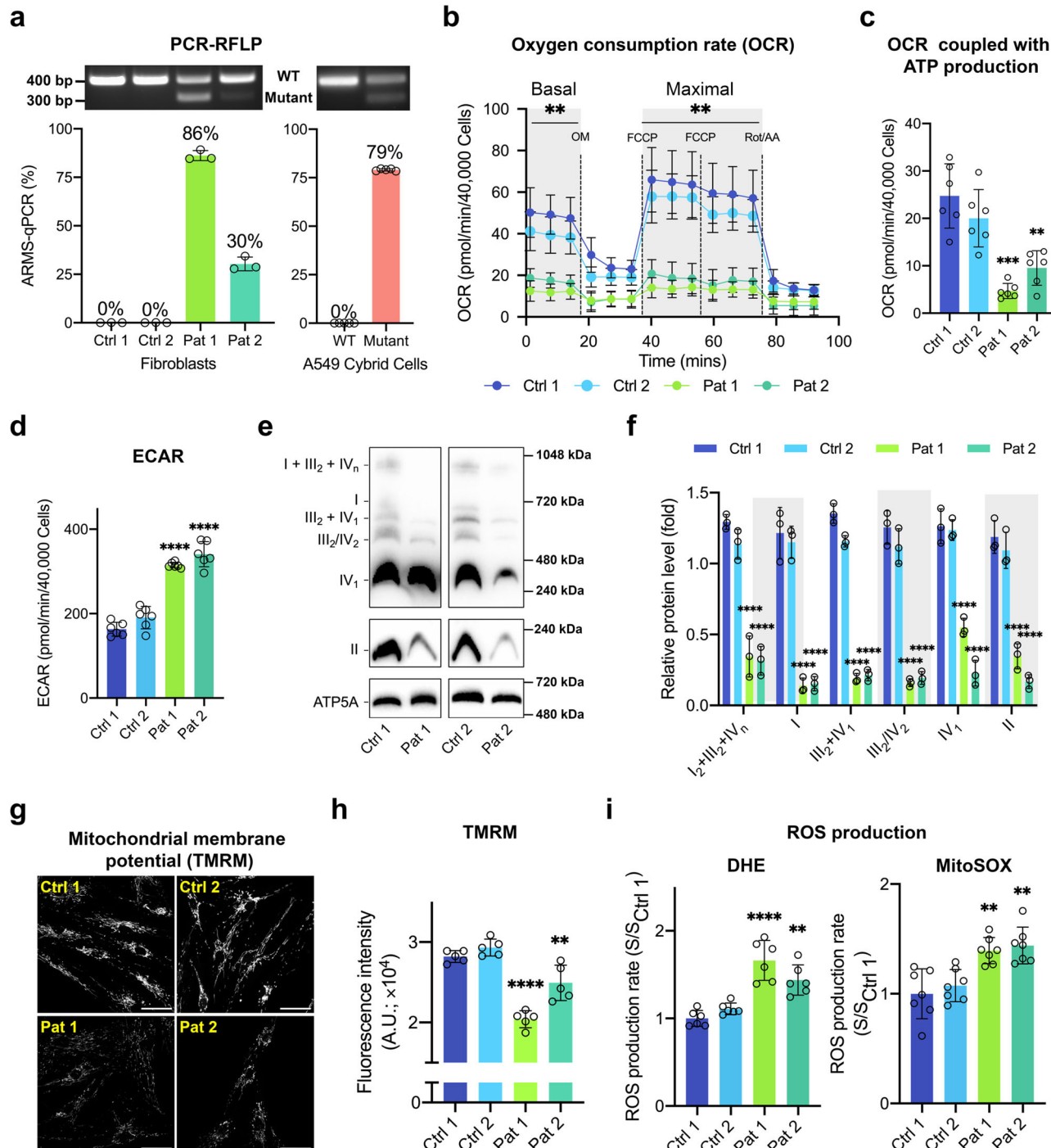

**Fig. 1 The m.3243 A > G mtDNA mutation causes mitochondrial dysfunction and oxidative stress. a** PCR-RFLP and ARMS-qPCR were used to quantify the mutation load for patient-derived fibroblasts ($n = 3$ independent biological samples) and A549 cybrid cells ($n = 5$ independent biological samples). **b**–**d** Cell respiratory capacity was measured using the Seahorse XFe96 extracellular flux analyser in patient fibroblasts ($n = 6$ culture wells) showing a major decrease in oxygen consumption under all conditions (**b**, $p < 0.0001$). Oxygen consumption dependent on ATP production (**c**, $p < 0.0001$)—the response to oligomycin—and extracellular acidification rate (ECAR) are plotted (**d**, $p < 0.0001$). **e**, **f** The expression of respiratory chain proteins and supercomplex assembly in patient fibroblasts were assessed using blue native gel electrophoresis (BNGE, **e**) and quantified (**f**, $p < 0.0001$), showing a major decrease in the assembly of all supercomplexes from the patient-derived cells ($n = 3$ independent experiments). **g**, **h** The mitochondrial membrane potential of fibroblasts was measured using TMRM with confocal microscopy (**g**) and quantified (**h**, $p < 0.0001$), showing a significant decrease in potential in patient-derived cells ($n = 5$ independent experiments). Scale bar = 50 μm. **i** ROS production rates of control and patient fibroblasts were measured using the reporters, DHE ($p < 0.0001$) and MitoSOX ($p < 0.0001$). Rates of ROS production were significantly increased in patient-derived cells ($n = 6$ independent biological samples). Source data are provided as a Source Data file. All data were represented as mean ± SD and data were analysed by one-way ANOVA with Tukey's multiple comparisons test (*$p < 0.05$, **$p < 0.01$, ***$p < 0.001$, ****$p < 0.0001$).

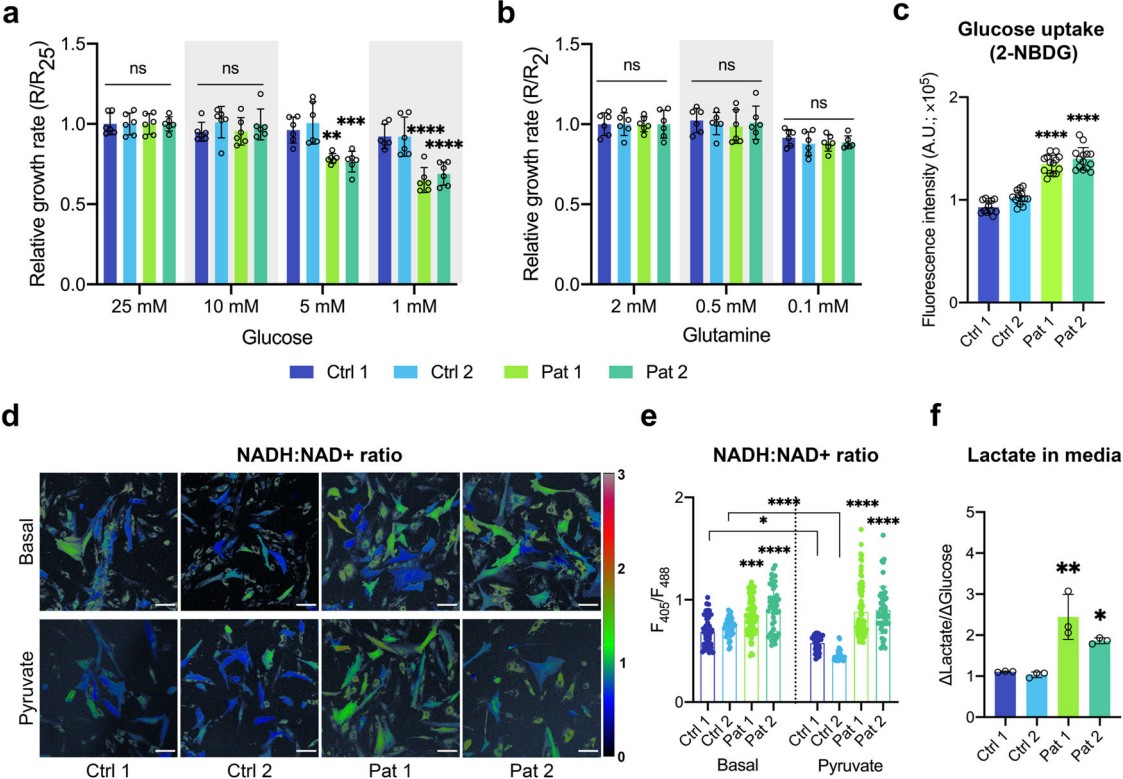

**Fig. 2 The m.3243 A > G mtDNA mutation switches cell metabolism to a more glycolytic phenotype, resulting in redox imbalance. a, b** The growth rates of patient fibroblasts were measured under a range of different nutrient conditions (normalised to the growth rate of each cell line in regular cell media), showing a decrease of growth rate compared to controls at glucose concentrations of 5 and 1 mM (**a**, $p < 0.0001$) but not at low glutamine (**b**) concentrations ($n = 6$ culture wells for all conditions, three independent experiments). **c** Glucose uptake in patient fibroblasts was measured using the fluorescent glucose analogue, 2-NBDG, showing a significantly increased rate of glucose uptake in the patient fibroblasts ($n = 14$ culture wells, $p < 0.0001$). **d, e** NADH:NAD+ ratio of fibroblasts was measured using the reporter, SoNar, under basal conditions ($n = 48, 46, 83$ and $46$ cells for Ctrl 1, Ctrl 2, Pat 1 and Pat 2 respectively) and after addition of pyruvate (200 μM, 30 min; $n = 42, 33, 83$ and $50$ cells for Ctrl 1, Ctrl 2, Pat 1 and Pat 2 respectively; **d**) and quantified (**e**, $p < 0.0001$). Scale bar = 100 μm. **f** Lactate production (normalised to glucose consumption) was measured in the media of fibroblasts using CuBiAn and showed a significant increase in lactate release in the media of patient-derived cells ($n = 4$ independent biological samples, $p = 0.0008$). Source data were provided as a Source Data file. All data were represented as mean ± SD and data were analysed by one/two-way ANOVA with Tukey's multiple comparisons test (\*$p < 0.05$, \*\*$p < 0.01$, \*\*\*$p < 0.001$, \*\*\*\*$p < 0.0001$).

exchange NADH and NAD+ between cytosol and mitochondrial matrix. We used the genetically encoded reporter SoNar[11] to quantify the cytosolic NADH:NAD+ ratio (Supplementary Fig. 2c). The basal NADH:NAD+ ratio was significantly increased in the patient fibroblasts compared to controls, consistent with increased regeneration of NADH through LDH activity (Fig. 2d, e). The addition of exogenous pyruvate can either support mitochondrial activity, increase OxPhos or stimulate LDH activity to generate lactate, changing the cytosolic NADH:NAD+ ratio. In control cells, pyruvate addition decreased the cytosolic NADH:NAD+ ratio, but had no significant effect on the high NADH:NAD+ ratio in the mutant cells (Fig. 2d, e), suggesting that in the latter, LDH flux is already saturated. These data are also consistent with significantly higher levels of lactate produced by the patient fibroblasts compared to controls (Fig. 2f). The pH of the growth media, reflecting cellular lactate production, was also measured using the absorbance of the pH indicator, phenol red. Of note, increased glucose uptake and dependence and imbalance of the cytosolic NADH:NAD+ ratio were recapitulated in the A549 cybrid cells (Supplementary Fig. 2d–h). The acidification of media by mutant cells was significantly greater than that of controls (Supplementary Fig. 2i). These data recapitulate the lactic acidosis which is characteristic of patients with the m.3243 A > G mutation[12], and suggest that in patient fibroblasts, increased activity of LDH may compensate for the decreased regeneration of

NAD+ by mitochondria, supporting increased glycolytic flux. Together, these data confirm that glucose uptake and glucose catabolism into lactate are upregulated and the cytosolic NADH:NAD+ ratio is increased in the mutant cells.

**The m.3243 A > G mutation remodels glucose metabolism towards increased lipid biosynthesis.** To investigate the glucose-dependent metabolic alterations in the mutant cells and identify metabolic signals which might drive changes in cell signalling, glucose uniformly labelled with isotopic carbon ([U-13C]-glucose) was used to trace its metabolic fate in the cells by gas chromatography–mass spectrometry (GC–MS). The pattern and abundance of 13C enrichment in downstream metabolites were used to determine the relative contribution of glucose into different metabolic pathways. First, partial least squares discriminant analysis (PLS-DA) for the concentration of metabolites (Fig. 3a) showed a clear separation between patient fibroblasts and controls (Supplementary Fig. 3a). The analysis of metabolic pathways revealed enrichment of glycerolipid metabolism, pentose phosphate pathway (PPP), glycolysis, the tricarboxylic acid (TCA) cycle and aspartate metabolism (Supplementary Fig. 3b). Tracing the fate of 13C-glucose through glycolysis (Fig. 3b), we found that the enrichment of 13C in glucose-6-phosphate (G6P) in the patient fibroblasts was at a similar level as in controls, but the total pools of G6P were increased (Fig. 3a) consistent with

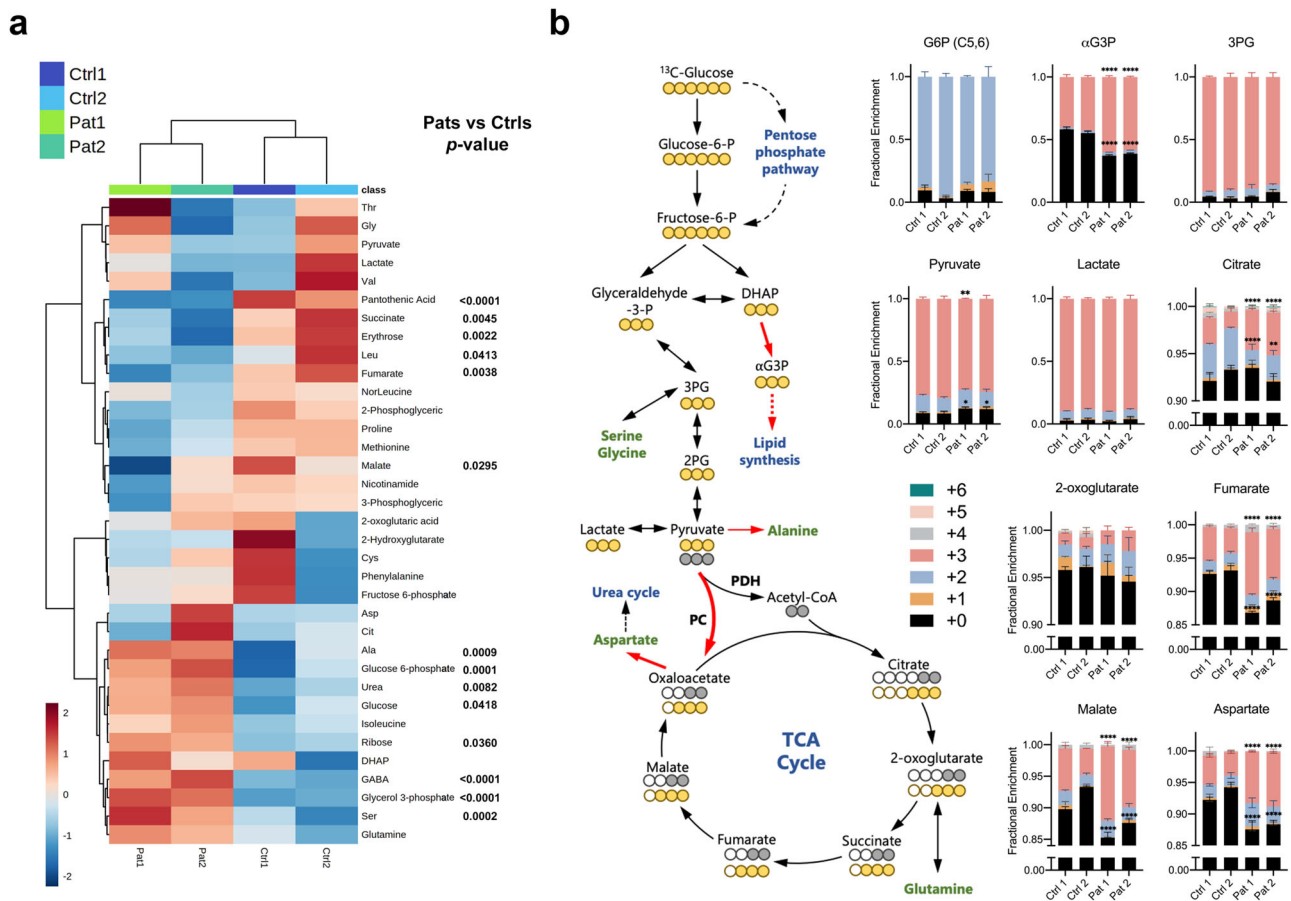

**Fig. 3 The m.3243 A > G mutation rewires glucose metabolism towards increased anabolic biosynthesis. a** The heatmap of metabolite concentrations obtained by GC–MS was generated by MetaboAnalyst 5.0 (*n* = 3 technical replicates), revealing significant differences in the concentrations of several metabolites between controls and patient fibroblasts (unpaired *t*-test; *p* values are shown). **b** Schematic of simplified glucose metabolism of the cells (Red arrows, upregulated pathways or reactions; three yellow circles of pyruvate and beyond, pyruvate is converted to OAA by pyruvate carboxylase; three or two grey circles of pyruvate and beyond, pyruvate is converted to acetyl-CoA by PDH. Tracing the distribution and abundance of $^{13}$C enrichment in the intermediates of glycolysis and the TCA cycle combined with Fig. 3a and Supplementary Fig. 3b revealed an increase in lipid synthesis and PC activity (*n* = 3 independent biological samples). G6P glucose-6-phosphate, αG3P glycerol-3-phosphate (*p* < 0.0001 in m + 0 and m + 3), 3PG 3-phosphoglyceric acid. Pyruvate, *p* = 0.0201 in m + 0 and *p* = 0.0082 in m + 3; Citrate, *p* = 0.0002 in m + 2 and *p* < 0.0001 in m + 3; Fumarate, *p* < 0.0001 in m + 0 and m + 3; Malate, *p* < 0.0001 in m + 0 and m + 3; Aspartate, *p* < 0.0001 in m + 0 and m + 3. Source data are provided as a Source Data file. All data, except Fig. 3a, are represented as mean ± SD and were analysed by one-way ANOVA with Tukey's multiple comparisons test for fibroblasts (**p* < 0.05, ***p* < 0.01, ****p* < 0.001, *****p* < 0.0001).

increased glucose uptake. Although the end products of glycolysis, intracellular pools of pyruvate and lactate in the patient fibroblasts, were maintained at similar levels with controls in terms of $^{13}$C enrichments and concentrations (Fig. 3a, b), the levels and the $^{13}$C enrichment of glycerol-3-phosphate (αG3P) were increased. The latter suggests the increased activity of αG3P dehydrogenase in response to increased cytosolic NADH:NAD$^+$ ratio, which may drive elevated lipid biosynthesis. Interestingly, while the absolute concentrations of alanine were increased in the mutant cells, its $^{13}$C enrichment was decreased, suggesting a contribution from other nutrient sources. Similarly, the total concentration of serine was increased in patient fibroblasts, despite decreased synthesis of serine from glucose, suggesting increased serine uptake from extracellular media (Fig. 3a, b and Supplementary Fig. 3c).

Consistent with OxPhos defects, levels of the TCA cycle intermediates, malate, fumarate and succinate were decreased in the mutant cells (Fig. 3a). Interestingly, the enrichment of m + 3 isotopologues of these TCA cycle intermediates and aspartate were significantly increased in the patient fibroblasts (Fig. 3b). To

enter the TCA cycle, glucose-derived pyruvate is converted into acetyl-CoA by pyruvate dehydrogenase (PDH). In this case the catabolism of [U-$^{13}$C]-glucose results in m + 2 enrichment of TCA cycle intermediates. In contrast, m + 3 isotopologues of TCA cycle intermediates are generated if pyruvate is converted into oxaloacetate (OAA), a precursor of aspartate, by pyruvate carboxylase (PC). Immunoblotting for PC and phospho-PDH expression revealed elevated PC expression in both patient fibroblasts and cybrid cells and increased levels of phosphorylated PDH (indicating its reduced activity) in patient 2 fibroblasts and cybrid cells (Supplementary Fig. 3d–f). As the glucose contribution into the TCA cycle is usually low in conventional cell culture systems, this slight but significant difference in the enrichment of $^{13}$C is meaningful. The switch from PDH to PC as means of channelling glucose-derived carbon into the TCA cycle in mutant fibroblasts is consistent with the increased NADH:NAD$^+$ ratio, which stimulates the activity of pyruvate dehydrogenase kinase, phosphorylating PDH and inhibiting its activity. Our findings from the stable-isotope-labelling approach and comparative metabolomics thus reveal a profound remodelling of glucose

metabolism as a consequence of OxPhos defects arising from the m.3243 A > G mutation.

**Chronic activation of the PI3K-Akt-mTORC1 axis in the m.3243 A > G mutant cells**. The metabolic phenotype shown above points to reprogramming of nutrient-sensing signalling networks in the m.3243 A > G mutant cells. We, therefore, carried out an RNA-sequencing screen to examine global changes in the expression profiles of the patient fibroblasts (Supplementary Fig. 4a). Using a significance level of false discovery rate (FDR) <0.05, we identified 3394 transcripts of which 1849 were upregulated and 1545 were downregulated (Supplementary Fig. 4b). A multidimensional principal component analysis also confirmed good accordance between biological triplicates as well as close clustering of transcripts from patient fibroblasts (Supplementary Fig. 4c). The enrichment analysis identified 'mitochondrial dysfunction', 'TCA cycle' as the altered metabolic pathways in patient fibroblasts (Supplementary Fig. 4d). Analysis of individual genes involved in these pathways showed a general downregulation of OxPhos genes (Supplementary Fig. 4e). Specifically, the increased expression of PC was consistent with the immunoblotting data (Supplementary Fig. 3d–f) and correlated with altered TCA cycle activity (Supplementary Fig. 4f). The enhancement of glycolytic flux in the mutant cells reported by the metabolomic data was supported by the increased expression of hexokinase 1 (HK1). A general decrease of enzymes in serine/glycine metabolism (Supplementary Fig. 4g) and an altered expression of amino acid transporters (Supplementary Fig. 4h) also matched our observation of increased serine in the metabolomics measurements. These findings further validated the metabolic footprint observed in patient-derived cells from the metabolomic studies.

Consistent with the above results, the network analysis of these differentially expressed genes identified the strong enrichment of PI3K-Akt and mTOR pathways in patient fibroblasts (Fig. 4a, Supplementary Fig. 4i, and Supplementary Table 1). We, therefore, validated these data at the protein expression level. Immunoblotting confirmed increased phosphorylation of Akt (S473) relative to total Akt, as well as phosphorylation of the downstream mTORC1 target S6 ribosomal protein in patient fibroblasts (Fig. 4b, c) and the cybrid cells (Supplementary Fig. 5a, b). While control fibroblasts showed a growth factor-dependent activation of Akt in response to increased serum concentration in growth media, phospho-Akt was elevated in patient cells even when growth factors were low (1% FBS) and increased only slightly in response to an elevated serum concentration. Similarly, the patient fibroblasts exhibited increased mTORC1 activation independently of the serum concentration, revealed by increased phospho-S6 levels. PI3K-Akt-mTORC1 signalling also antagonises the activity of the cellular energy sensor, AMP-activated protein kinase (AMPK) and suppresses macromolecular catabolism by autophagy. We, therefore, measured AMPK phosphorylation in response to different serum concentrations by immunoblotting but found no significant difference between patient and control fibroblasts. We further measured phosphorylation of the Akt substrates, p-TSC2 (S939) and p-PRAS40 (T246), and the mTORC1 substrate, p-4EBP1 (S65), of patient fibroblasts grown in 10 or 1% FBS media. In all of these, phosphorylation was increased in cells carrying the mtDNA mutation independent of serum concentration, again consistent with constitutive activation of the pathway in the mutant cells (Supplementary Fig. 5c). These results suggest that the constitutive induction of PI3K-Akt-mTORC1 signalling remodels cell metabolism, independently of growth factor stimulation.

To establish that the PI3K-Akt-mTORC1 axis is also constitutively active in patients with the m.3243 A > G mtDNA mutation, we measured p-Akt/Akt and p-S6/S6 by immunofluorescence in a muscle biopsy from a patient with an 82% mutant load of the m.3243 A > G mutation causing MELAS, (not one of the donors used to obtain the fibroblasts; matched H&E and COX staining are shown in Supplementary Fig. 5d). These data confirmed a significant increase in p-Akt and p-S6 in the muscle biopsy, which argues that upregulation of this pathway is a clinical feature of the disease and not a function of the cultured cells (Fig. 4d, e). Reanalysis of a published RNA-seq data set[13] from muscle biopsies of MELAS patients also confirmed the activation of PI3K-Akt-mTORC1 signalling in patients (Supplementary Table 2). In summary, activation of the PI3K-Akt-mTORC1 axis is a common feature of m.3243 A > G mutation both in vivo and in vitro.

**Pharmacological inhibition of PI3K-Akt-mTORC1 signalling reduces mutant load, improves mitochondria function and lowers glucose dependence in the m.3243 A > G mutant cells**. At this point, we asked what the functional consequences of the activation of the PI3K-Akt-mTORC1 axis are and whether it represents an adaptation to impaired OxPhos. We therefore systematically inhibited each component of the axis, in turn, using well-established pharmacological inhibitors (Fig. 5a)[14] and explored the consequences for mutant mtDNA burden and the pathophysiological and biochemical features that we have described above.

These inhibitors include two inhibitors of class 1 PI3K, LY294002 (LY, 5 μM) and GDC0941 (GDC, 1 μM), an inhibitor of Akt, MK2206 (MK, 1 μM) and an inhibitor of mTORC1, rapamycin (RP, 5 μM). In both the fibroblasts from patient 1 and the cybrid cells, treatment with LY, GDC, MK or RP, led to a striking and progressive decrease in mtDNA mutant load over 6–12 weeks of sustained treatment in which the mutant load fell from ~80% to values as low as 40% (Fig. 5b). Interestingly, in the absence of the inhibitors, the fibroblasts from patient 2 showed a progressive increase in mutant load with time in culture, rising from ~30% to around 70% over 3 months (Fig. 5b). Treatment with any of the inhibitors significantly suppressed the rate of increase in mutant load decreasing the total change by around 20%. Although the decrease in mutant load took weeks to develop, changes in protein phosphorylation state were evident within a day of starting treatment (Supplementary Fig. 6a–g). Accordingly, inhibition of PI3K by LY or GDC or inhibition of Akt by MK2206 reduced Akt and mTOR phosphorylation, while rapamycin inhibited mTOR phosphorylation but had no impact on Akt. None of the treatments had any immediate effect on AMPK phosphorylation. However, AMPK activity gradually increased at later time points after drug treatments, suggesting induction of catabolic processes such as autophagy and improvement in sensing the bioenergetic status of the m.3243 A > G mutant cells[15].

To investigate whether the decrease in mutant load after drug treatment was reflected by a concomitant change in mitochondrial function, we measured the mitochondrial bioenergetic status and the metabolic phenotype of the patient-derived cells. The mitochondrial membrane potential of the cells treated with RP or LY showed complete recovery to levels that were not significantly different from control cells (Fig. 5c, d). Although $\Delta\psi_m$ of the cells treated with GDC or MK was higher than the untreated patient fibroblasts, it was still slightly but significantly lower than controls. Similarly, measurements of mitochondrial respiration showed that both basal (50–65% of controls) and maximal respiratory capacity (69–86% of controls) were increased in cells

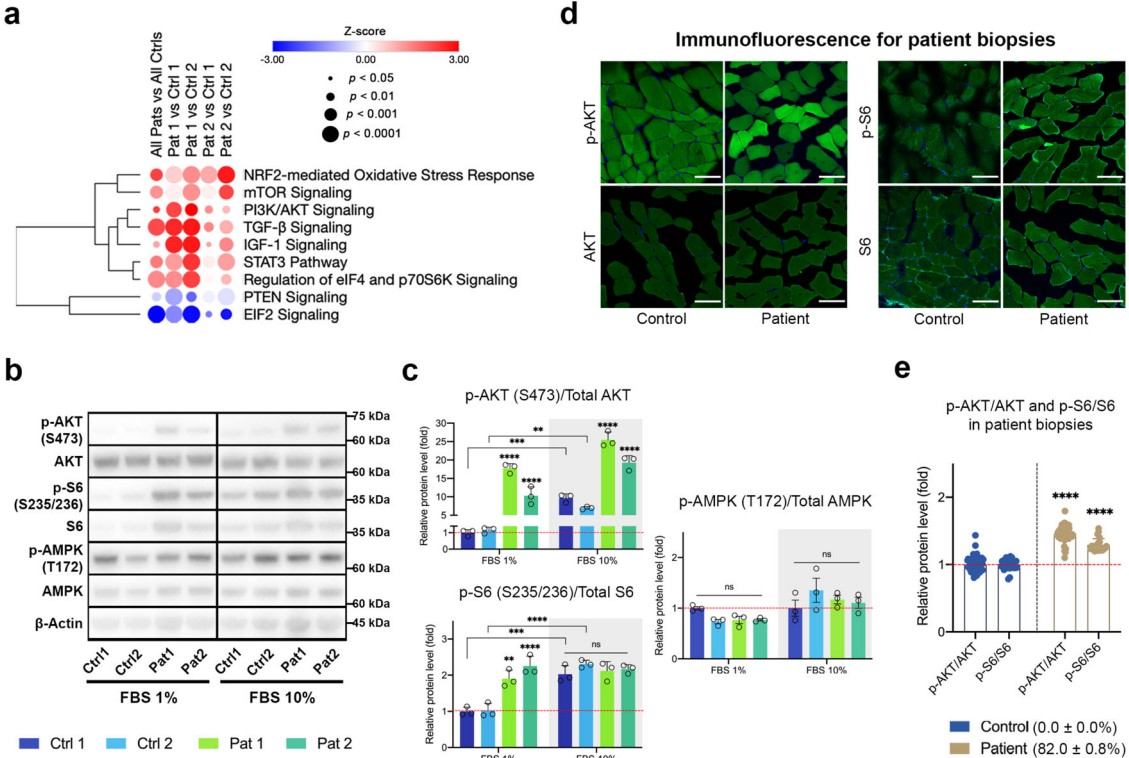

**Fig. 4 The PI3K-Akt-mTORC1 axis is upregulated in the m.3243 A > G mutant cells. a** Analysis of RNA-seq data from the patient fibroblasts by QIAGEN ingenuity pathway analysis (IPA) showed cell signalling pathways, which are enriched in cells carrying the m.3243 A > G mutation. These include striking and significant increases in the expression of pathways involving PI3K-Akt, mTOR, EIF2, and NRF2. It is notable also that PTEN expression, a negative regulator of PI3K-Akt-mTOR signalling was downregulated ($p < 0.05$). **b, c** Immunoblotting of p-Akt (S473)/Akt, p-S6 (S235/236)/S6 and p-AMPK (T172)/AMPK in patient fibroblasts grown in the presence of 10 or 1% FBS media (**b**), showing a major increase in p-Akt/Akt ($p < 0.0001$) and p-S6/S6 ($p < 0.0001$) in the presence of 1% FBS media in the mutant cells; in contrast, there is no difference in p-AMPK/AMPK (**c**; $n = 3$ independent experiments). **d, e** Immunofluorescence staining for p-Akt (S473)/Akt ($n = 66$ and 78 muscle fibres for Control and Patient, respectively; $p < 0.0001$) and p-S6 (S235/236)/S6 ($n = 30$ and 23 muscle fibres for Control and Patient, respectively; $p < 0.0001$) in patient muscle biopsies (**d**) confirmed the increased phosphorylation of Akt and S6 in the patient muscle biopsies (**e**). Scale bar = 100 μm. Source data are provided as a Source Data file. All data, except Fig. 4a, are represented as mean ± SD and were analysed by one-way ANOVA with Tukey's multiple comparisons test for fibroblasts and by unpaired t-test for biopsies (*$p < 0.05$, **$p < 0.01$, ***$p < 0.001$, ****$p < 0.0001$).

treated with any of the inhibitors (Fig. 5e). Of note, the improvement of mitochondrial bioenergetics by these inhibitors was recapitulated in the A549 cybrid cells (Supplementary Fig. 6h–j).

We then focused on RP and LY, the two most potent inhibitors, for further experiments. The improvement in mitochondrial respiration suggested a metabolic shift towards OxPhos. Measurements of cytosolic redox state using SoNar (Fig. 5f, g and Supplementary Fig. 6k, l) showed a significant reduction in basal NADH:NAD$^+$ ratio and increased response to pyruvate addition in the long-term treated patient-derived cells, implying a decreased glycolytic flux. Measurements of glucose and lactate concentrations in the media showed a profound decrease in lactate secretion following LY and RP treatments (Fig. 5h). The patient fibroblasts were also cultured in media containing glucose or galactose as a primary carbon source to further assess their dependence on glycolysis. LY- and RP-treated cells, but not mutant controls, grew well in the galactose media, indicating a reduced glycolytic dependence after the drug treatments (Fig. 5i). Also, the patient fibroblasts showed an increased supercomplex assembly as well as the partial rescue of individual components of the mitochondrial ETC following all drug treatments (Fig. 5j, k). These changes were accompanied by a decrease in the rates of ROS production in cells treated with LY and RP (Fig. 5l). These changes in cell metabolism were again recapitulated in the A549 cybrid cells (Supplementary Fig. 6m–p).

Altogether, these data demonstrate that the long-term treatment of the mutant cells with inhibitors of the PI3K-Akt-mTORC1 axis reduces mutant load and reverses the biochemical consequences of the mutation, improving mitochondrial function, reducing glucose dependence and lactate secretion. These data strongly argue that the chronic constitutive activation of the PI3K-Akt-mTORC1 axis serves as a maladaptive response in this disease model and helps to sustain the mtDNA mutant load.

**Reduction of the m.3243 A > G mutant load by inhibition of the PI3K-Akt-mTORC1 axis is cell-autonomous.** The chronic activation of the PI3K-Akt-mTORC1 axis and increased mutant load have deleterious effects on cellular function and amplify the pathophysiological consequences of the mutation. However, this raises questions about the underlying mechanism by which inhibition of the axis reduces mutant load and improves mitochondrial function. A possible explanation for the response to drug treatments is that inhibition of the signalling pathway may promote the selective growth of cell populations with low mutant load or selective elimination of cells with high mutant loads. We, therefore, carried out long-term measurements of growth rates and cell death (Fig. 6a). Cells were pretreated with either vehicle, LY or RP for the time periods as specified in Fig. 6b ('pretreatment') and then seeded into 96-well plates for measuring cell growth rate and cell death with either vehicle, LY or RP

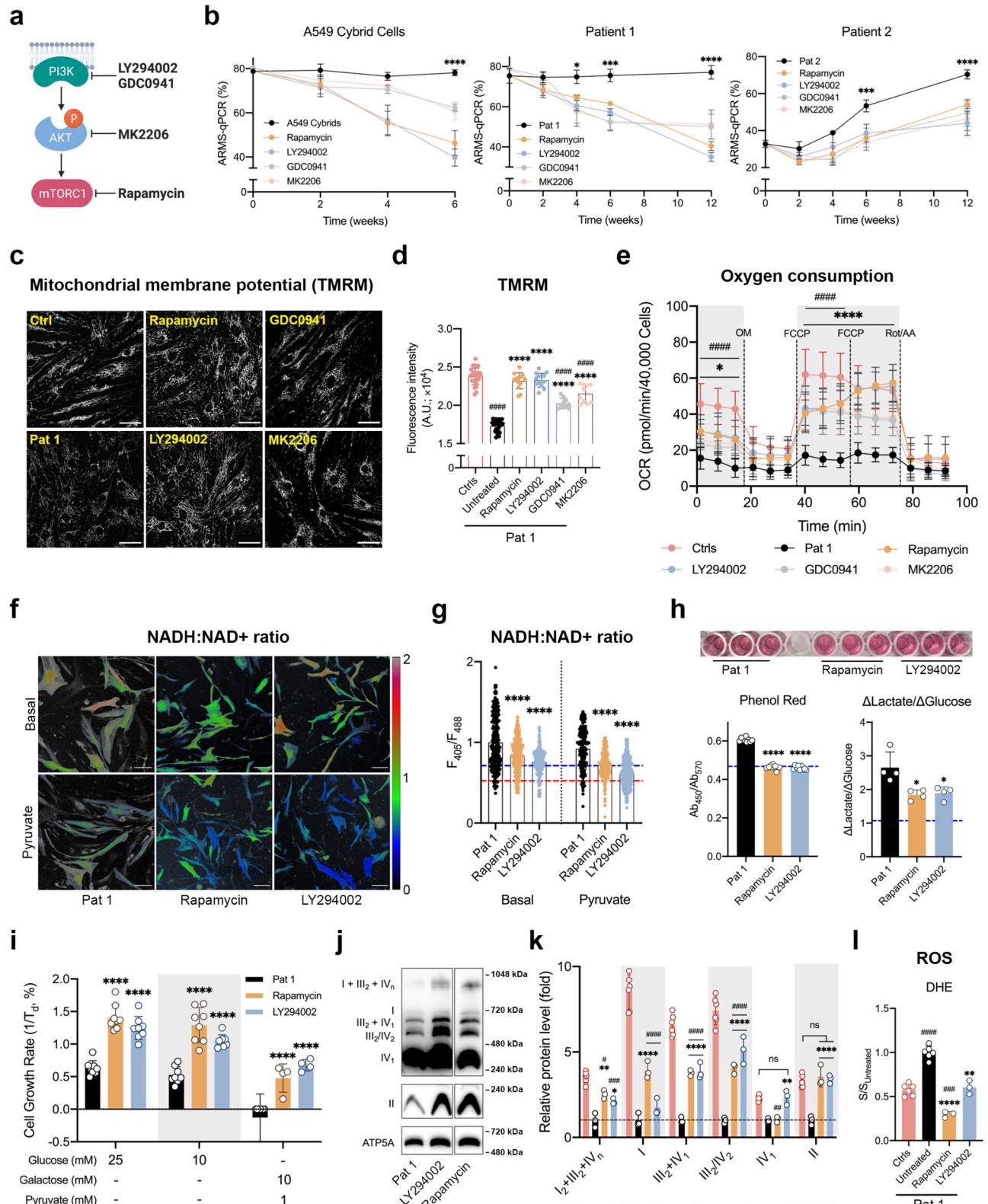

('treatment'). Although treatment with either LY or RP significantly slowed cell proliferation in both fibroblast and cybrid cells at the time point of 0-week pretreatment (Fig. 6b, upper panel), the changes were small. The cell growth rate was stable in cybrid cells treated with LY or RP, suggesting that there was no clonal expansion. In fibroblasts, growth rates of cells were not altered by LY but increased by RP after 4-week pretreatment.

However, the growth rate of patient 1 fibroblasts was not further increased by RP after 8-week pretreatment. In contrast, a significant decrease of mutation load was clearly established after 4 weeks of treatment, suggesting that the change of mutation load is independent of the rate of cell growth. On the other hand, cell death in the drug-treated groups were even lower than in the mutant controls, suggesting that the drug treatments were

**Fig. 5 Inhibitors of the PI3K-Akt-mTORC1 axis, LY294002 and Rapamycin, reduce mutation load, partially rescue mitochondrial function and reduce glucose dependence. a, b** Schematic depicting pharmacological inhibition of the PI3k-Akt-mTORC1 axis by LY290042 (LY), GDC0941 (GDC), MK2206 (MK) or Rapamycin (RP) (**a**; created with BioRender.com.). **b** Sustained treatment of cells over 6 or 12 weeks with LY (5 μM) GDC (1 μM), MK (1 μM) or RP (5 μM) caused a progressive decrease in relative mutant mtDNA load in A549 cybrid cells and fibroblasts of patient 1 and suppressed the progressive increase of mutant load with time in culture seen in patient 2 fibroblasts ($n = 3$ independent experiments; $p < 0.0001$ in all three cell lines). The concentrations here refer to all subsequent treatments. **c, d** The mitochondrial membrane potential (TMRM 25 nM) of patient 1 fibroblasts was significantly increased following treatment with LY, GDC, MK or RP for 12 weeks ($n = 10$ independent experiments; $p < 0.0001$). Scale bar = 50 μm. **e** The respiratory capacity of patient 1 fibroblasts treated with LY, GDC, MK or RP for 12 weeks showed a major increase in oxygen consumption under all conditions following the drug treatments cells ($n = 14$ of culture wells; $p < 0.0001$). **f, g** The NADH:NAD$^+$ ratio of patient 1 fibroblasts treated with LY or RP for 12 weeks was measured under basal conditions ($n = 243$, 280 and 285 cells for Pat 1, RP and LY, respectively) and in the presence of pyruvate (200 μM, 30 min; $n = 188$, 272 and 342 cells for Pat 1, RP and LY, respectively; **f**) and quantified (**g**; $p < 0.0001$). The data showed a significant decrease compared to pretreatment levels (blue dash line, basal NADH:NAD$^+$ ratio of controls; red dash line, NADH:NAD$^+$ ratio of controls under pyruvate condition). Scale bar = 25 μm. **h** Acidification of the growth medium was significantly reduced following chronic drug treatments with RP and LY ($n = 9$ culture wells; $p < 0.0001$). Lactate production ($n = 4$ culture wells; $p = 0.0070$) was also significantly reduced. Blue dashed line, basal levels of controls. **i** Fibroblasts of patient 1 treated with LY or RP for 12 weeks were also cultured in media with a variety of glucose/galactose concentrations to further assess the glucose dependence, displayed a significant improvement of cell growth in all conditions from the drug-treated cells ($n = 8$ culture wells for the 25 and 10 mM glucose groups and $n = 4$ culture wells for the galactose + pyruvate group; $p < 0.0001$). **j, k** Respiratory chain proteins and supercomplex assembly of patient 1 fibroblasts treated with LY or RP (**j**) showed a major and significant increase in the assembly of almost all supercomplexes from the drug-treated cells (**k**, $n = 3$ independent experiments; $p < 0.0001$). **l** ROS production rates of patient 1 fibroblasts treated with LY or RP for 12 weeks were significantly reduced to the level that was lower than or no difference from controls ($n = 6$ wells for Ctrls and Pat 1; $n = 3$ wells for the RP and LY groups; $p < 0.0001$). Source data are provided as a Source Data file. All data represented as mean ± SD and were analysed by one/two-way ANOVA with Tukey's multiple comparisons test (*$p < 0.05$, **$p < 0.01$, ***$p < 0.001$, ****$p < 0.0001$, vs patient/mutant control; #$p < 0.05$, ##$p < 0.01$, ###$p < 0.001$, ####$p < 0.0001$, vs WT control).

beneficial, and there was no selective killing of mutant cells (Fig. 6b, lower panel). Treating the cybrid cells with GDC or MK showed similar results in terms of cell growth and death (Supplementary Fig. 7a, b).

To determine definitively whether the reduction of mutant load in response to the drug treatments is cell-autonomous, we used a PCR-based technique, TaqMan SNP genotyping, at single-cell resolution to measure the distributions of the m.3243 A > G heteroplasmy in a cell population (Fig. 6a). The TaqMan technique was first validated by measuring the mutation loads of a population of cells and then to establish the range of single-cell mutant load across the population (Supplementary Fig. 7c). The distribution of mutant load in the cybrid cells ranged from 50–80%, showing a Gaussian distribution (Supplementary Fig. 7d). Although we found that this technique is less accurate than ARMS-qPCR and tends to overestimate the mutant load when it is lower than 50% and underestimate the mutant load when it is higher than 50%, the average single-cell mutant load was consistent with that of the whole cell population, suggesting a good precision of the technique. Measurement of the relative m.3243 A > G heteroplasmy in patient fibroblasts and cybrid cells following treatment with LY and RP for 6 and 12 weeks respectively showed a frequency distribution, which clearly segregated the treated group as a distinct population from the untreated group (Fig. 6c, d and Supplementary Fig. 7e, f). The distribution of the mutant load in treated groups shifted to significantly lower levels compared to the untreated group.

To determine whether the reduction in the m.3243 A > G mutant load is phenocopied at the protein expression level, we immunostained the mtDNA-encoded cytochrome c oxidase I (MT-COI) in patient 1 fibroblasts. The presence of 7 UUA-encoded leucine residues within MT-COI renders the m.3243 A > G tRNA$^{Leu(UUR)}$ cells prone to a codon-specific translational defect[16]. Immunofluorescence staining revealed a significant decrease in MT-COI expression in patient 1 fibroblasts compared to controls (Supplementary Fig. 7g–i). Following drug treatments, these cells showed a partial but significant recovery of MT-COI expression. The single-cell analysis of MT-COI expression segregated the drug-treated and untreated groups in a distribution pattern similar to the m.3243 A > G mtDNA distribution

shown in Fig. 6d. A diminished expression of MT-COI also correlated with the reduced assembly and activity of super-complexes shown in Fig. 5k. Together, these results strongly suggest that rather than the enrichment of a specific cell population, the reduction in m.3243 A > G mtDNA mutant load caused by inhibition of PI3K-Akt-mTORC1 signalling operates across the whole population and is cell-autonomous.

## Discussion

The clinical presentation of diseases caused by pathogenic mtDNA mutations is highly heterogeneous[4–6]. The association between specific mtDNA mutations, heteroplasmic mutant load, disease manifestation and severity are poorly understood. In the present study, we have systematically characterised the metabolic and cell signalling phenotype of cells bearing the m.3243 A > G mutation, the most prevalent mtDNA mutation, which is responsible for mitochondrial encephalomyopathy, lactic acidosis and stroke-like episodes (MELAS) or maternally inherited diabetes and deafness (MIDD) and related disorders. We found that the changes in metabolism were associated with the constitutive hyperactivation of the PI3K-Akt-mTORC1 pathway specifically to the m.3243 A > G mtDNA mutation. Most notably, pharmacological inhibition of PI3K, Akt, or mTORC1 in the patient-derived cells reduced mutant load and partially rescued mitochondrial bioenergetic function (Fig. 7). In contrast to the m.3243 A > G mtDNA variant, previous studies have reported downregulation of Akt activity in cells carrying the m.8993 T > G mutation[17]. To confirm that the mechanisms are specific to the m.3243 A > G mutation, we also examined the effects of LY and RP in cells carrying another heteroplasmic mtDNA point mutation—the m.8993 T > G—generated by the Minczuk lab[18]. In these cells, there was no difference in the phosphorylation of Akt and S6; furthermore, inhibition of the PI3K-Akt-mTORC1 axis had no impact on mutant load (Supplementary Fig. 7j, k), suggesting that hyperactivation of the PI3K-Akt-mTORC1 axis is relatively disease specific. These findings suggest that the changes in signalling pathways are disease specific and may represent a link between genotype and phenotype.

Metabolic features of the patient fibroblasts and cybrid cells are consistent with most of the previous studies of the m.3243

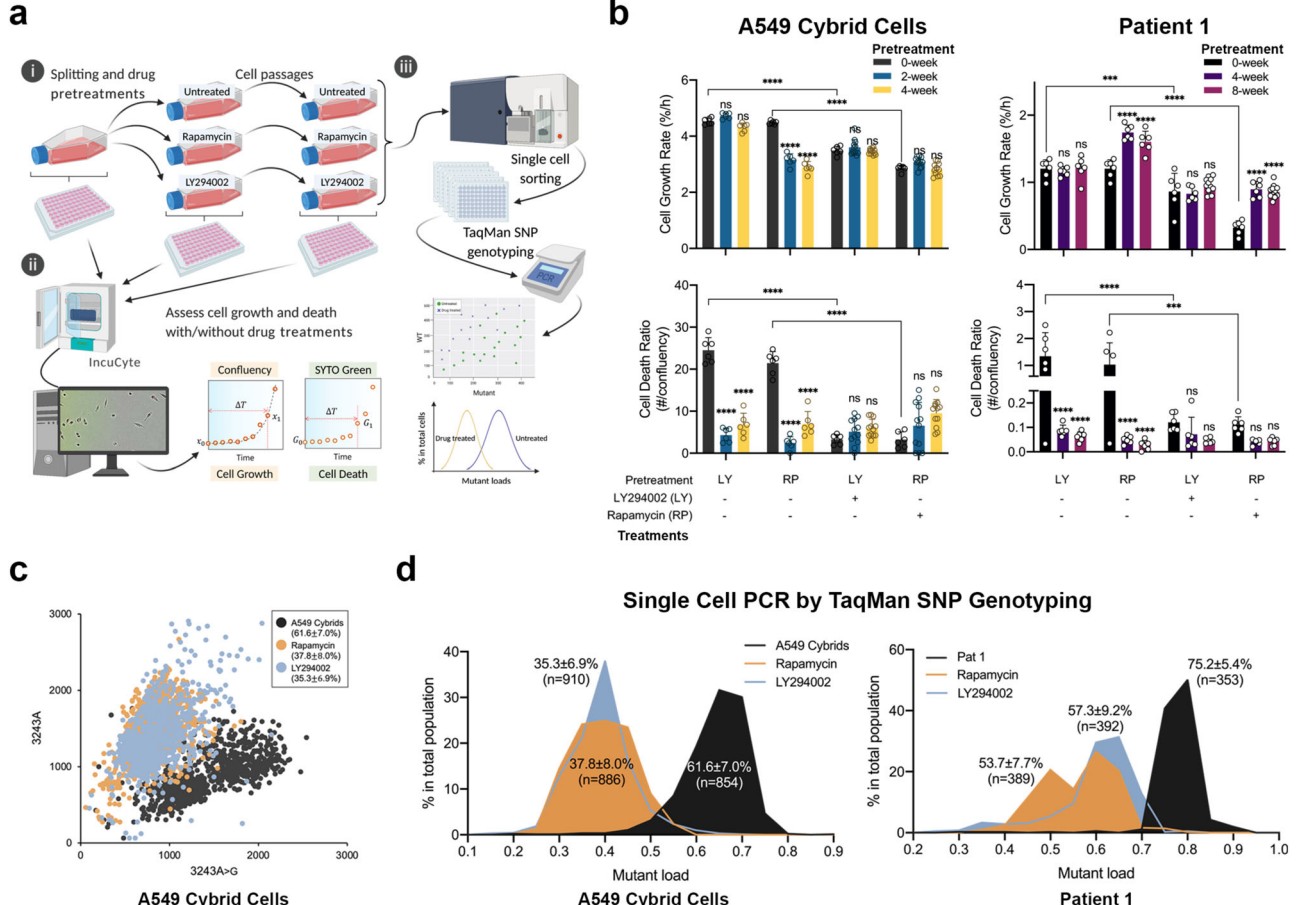

**Fig. 6 Reduction of the m.3243 A > G mutant load by inhibition of PI3K-Akt-mTORC1 requires neither selective cell death nor clonal expansion and is a cell-autonomous event. a** A flow chart describing the cell culture process used to assess cell growth/death over the 4- or 8-week drug pretreatments. Cells were pretreated with either vehicle, LY or RP for the time periods as specified in Fig. 6b ('pretreatment') and then seeded into 96-well plates for measuring cell growth/death with either vehicle, LY or RP ('treatment'). Single-cell PCR for the m.3243 A > G mutant loads was performed at the end of pretreatments using TaqMan SNP genotyping. The chart was created with BioRender.com. **b** Cell growth (upper panel) and cell death (lower panel) over the 4- or 8-weeks drug pretreatments of A549 cybrid cells (left panel) and patient 1 fibroblasts (right panel) were measured using Incucyte ($n = 6$ culture wells; $p < 0.0001$). Although treatment with either LY or RP at 5 μM slowed cell proliferation in both patient 1 fibroblasts and cybrid cells, the cell death numbers were also lowered by the drug treatments. **c, d** A scatter plot showing the distribution of WT and mutant mtDNA measured in single A549 cybrid cells treated with LY or RP for 6 weeks showed a clear shift of mutant load distribution (**c**). **d** Mutant load distributions of single A549 cybrid cells ($n > 850$ cells) and patient 1 fibroblasts ($n > 350$ cells) treated with LY or RP for 6 and 12 weeks, respectively. Source data are provided as a Source Data file. All data represented as mean ± SD and were analysed by one/two-way ANOVA with Tukey's multiple comparisons test (*$p < 0.05$, **$p < 0.01$, ***$p < 0.001$, ****$p < 0.0001$).

A > G mutation, including mitochondrial dysfunction, upregulated glycolysis, increased ROS production and redox imbalance[5,12,15,16,19,20]. However, metabolic profiling of cells bearing the mutation is limited in the literature. Metabolomics data showed that reductive carboxylation of glutamine is a major mechanism to support cell survival and maintain redox balance in the m.8993 T > G mutation cell models[18,21]. In contrast, using a similar approach, we found that glucose anabolism (i.e. upregulated upper glycolysis, PPP and lipid synthesis) is increased in the m.3243 A > G mutant cells and these cells are dependent on glucose for cell survival and proliferation. Also, increased TCA cycle entry of pyruvate through PC, which, combined with the prediction of enriched malate-aspartate shuttle pathway (Supplementary Fig. 3b) may serve as a mechanism to maintain cellular NADH:NAD$^+$ balance to support glycolytic flux and promote antioxidant capacity[18,22]. This difference in metabolism between the m.8993 T > G and m.3243 A > G mutations argues that the mtDNA mutations cannot be seen as one disease[23,24]. Thus, the rewired metabolism specific to the m.3243 A > G

mutation drew our attention to the corresponding changes in cell signalling.

The alterations of metabolism in the m.3243 A > G mutant cells strongly imply a perturbation in cell signalling, especially implicating the PI3K-Akt-mTORC1 pathway, which has been widely studied in cancer signalling and metabolism[25]. For example, oxidative stress and redox imbalance may activate PI3K-Akt, the increased glycolytic flux and PPP implies an activated Akt-NRF2 signalling, the increased PC expression and p-PDH suggests an accumulation of acetyl-CoA associated with Akt-mTORC1 activity[25]. Metabolic pathway analysis using RNA-seq data recapitulated these metabolic changes in the patient fibroblasts. Moreover, network analysis using the RNA-seq data further validated the link between the metabolic changes and altered cell signalling in cells carrying the m.3243 A > G mutation.

Crosstalk between the PI3K-Akt-mTORC1 axis and mitochondrial quality control pathways, including mitochondrial fusion-fission dynamics, mitochondrial biogenesis and mitophagy, is critically involved in regulating mitochondrial energy

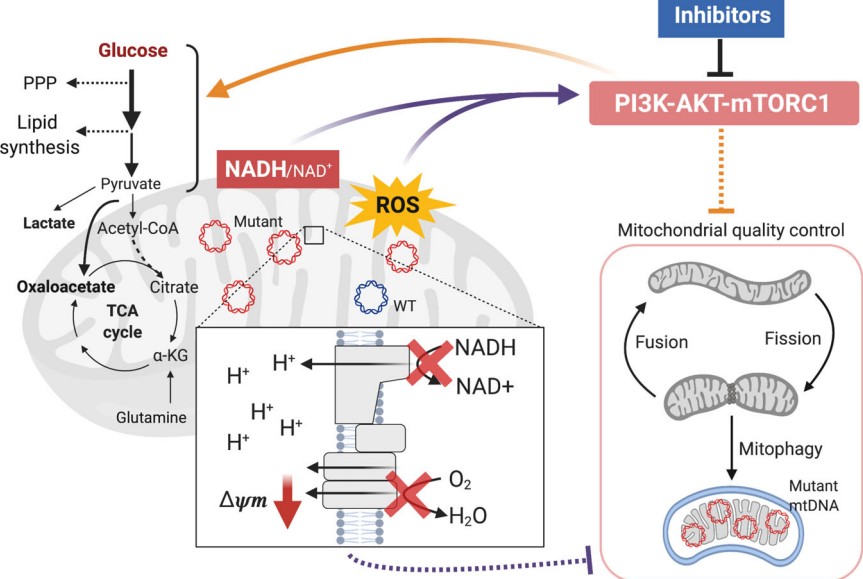

**Fig. 7 Inhibition of the PI3K-Akt-mTORC1 axis reduces the m.3243 A > G mutant load.** A scheme describing how the m.3243 A > G mutation alters cell metabolism, including mitochondrial dysfunction, altered glucose metabolism, redox imbalance and oxidative stress, which may lead to the constitutive activation of the PI3K-Akt-mTORC1 pathway. Pharmacological inhibition of this pathway reduces mutant load, partially rescues mitochondrial function and reduces glucose dependence in a cell-autonomous way. Although the underlying mechanisms for selecting against the mutant mtDNA need further investigations, these findings suggest that the activation of the PI3K-Akt-mTORC1 pathway drives a positive feedback cycle, which maintains/increases mutant mtDNA, augments the metabolic rewiring and thus exacerbates the perturbation of cell signalling. This positive feedback loop may shape the phenotype of the disease and determine disease progression. The scheme was created using BioRender.com.

homoeostasis[26,27]. Several studies have demonstrated the accumulation of damaged mitochondria and defective mitophagy as a hallmark of mtDNA diseases and age-related neurodegeneration, suggesting that the presence of pathogenic mtDNA alone is not sufficient to drive selection against the mutation by the activation of mitophagy[19,28–31]. It is still unclear whether chronic activation of PI3K-Akt-mTORC1 signalling impairs the autophagy/mitophagy pathway, leading to accumulation of damaged mitochondria in the m.3243 A > G mutant cell. On the other hand, the single-cell analysis (Fig. 6) strongly argues that the change in mutant load by the inhibition of PI3K-Akt-mTORC1 axis is a cell-autonomous phenomenon. A possible explanation could be the elimination of the mutant mtDNA by reducing the total mtDNA copy number while amplifying the dominant mtDNA, resembling 'the bottleneck' effect—the selection mechanism of mtDNA in the germline or embryo[4,6]. It also seemed plausible that the removal of dysfunctional mitochondria via mitophagy may reduce the mutant load in cells with the mtDNA mutation[32,33]. Further investigations for the mechanisms selecting against the mutant mtDNA in our cell models are needed.

Indeed, the network analysis also identified eukaryotic initiation factor 2 (EIF2) and phosphatase and tensin homologue (PTEN) signalling as the top downregulated pathways (Fig. 4a). PTEN is a negative regulator of the PI3K-Akt-mTORC1 axis. Although the mechanisms of crosstalk between PI3K-Akt-mTORC1 and EIF2 pathways are still unclear, these two pathways seem to communicate and determine cell fate under stress[34,35]. Of note, IPA upstream analysis showed an upregulation of transcription factors in the EIF2 pathway—ATF4 and DDIT3 (CHOP) (Supplementary Tables 2, 3). ATF4 acts as a prototypical downstream target coupling mitochondrial proteotoxic stress to the activation of integrated stress response (ISR)[36] and derives remodelling of one-carbon metabolism (serine/glycine and folate metabolism)[37]. Although we observed an increase

of serine concentration in fibroblasts bearing the m.3243 A > G mutation, incorporation of $^{13}$C-glucose into serine was reduced. Also, expression of genes related to serine/glycine metabolism was generally reduced (see Supplementary Fig. 4g). These data imply that in fibroblasts carrying the m.3243 A > G mutation, unlike 'Deletor' mice[37], serine synthesis is downregulated. On the other hand, given that the mitochondrial unfolded protein response (UPR$^{mt}$) maintaining mutant mtDNA in a *C. elegans* model is mitopahgy independent[38], exploring the role of ISR/UPR$^{mt}$ in the maintenance of the m.3243 A > G mutation and disease progression by suppressing eIF2α phosphorylation will be interesting. Furthermore, in a published RNA-seq data set of muscle biopsies from patients with MELAS[13], FGF21 (~48-fold) and GDF15 (~9-fold) were upregulated, matching previous findings reported from muscle tissue of 'Deletor' mice[37]. In contrast, in the present study, while we did not find changes in these two genes in our RNA-seq data, we did find upregulation of FGF16 (~33-fold) in the patient fibroblasts, suggesting that changes in FGF pathways in response to mitochondrial dysfunction might be tissue specific.

Previous studies have suggested that different levels of heteroplasmy of the m.3243 A > G mutation in cybrid cells result in different gene expression patterns and even in discrete changes in metabolism[39,40]. Another study showed that JNK (c-Jun N-terminal kinases) was activated by ROS in cybrid m.3243 A > G cells, further reducing RXRA (retinoid X receptor alpha) expression[41]. In contrast, our data did not show inhibition of the RXRA pathway and suggest that chronic activation of the PI3K-Akt-mTORC1 pathway is a general response in cells with the m.3243 A > G mutation despite very different levels of mutant load. Remarkably, we found that prolonged treatment with inhibitors of each component of the PI3K-Akt-mTORC1 axis reliably and reproducibly reduced mutant burden and partially rescued mitochondrial function in the mutant cells. Although

inhibition of PI3K-Akt-mTORC1 axis has been shown to be beneficial in several mitochondrial and neurological disease models, the underlying mechanisms remain elusive[14,30,42–44]. As a result, the therapeutic efficacy of the inhibition may be distinct and limited in different models. A study of a mouse model of Leigh syndrome showed that rapamycin, while delaying the progression of the disease, failed to improve OxPhos[45]. Similarly, another study showed that mTORC1 inhibition failed to improve either mitochondrial bioenergetics or survival in a mouse model of mitochondrial encephalomyopathy (Coq9[R239X])[46]. Our data from cells with the m.8993 T > G mutation suggest that changes in signalling pathways may differ between mtDNA diseases, perhaps pointing towards the mechanisms that define differences in disease phenotype between different mitochondrial diseases, and emphasising that therapeutic options should be considered separately for each disease related to mtDNA mutations. However, since both the PI3K-Akt-mTORC1 axis and the accumulation of mtDNA mutations have been associated with neurodegeneration, ageing and cancers, our study may implicate in a broader biomedical context[4,6,14,25,26].

Overall, the m.3243 A > G mutation causes a profound cell metabolic and signalling remodelling, although the causality between the observed alterations in metabolism and cell signalling of the m.3243 A > G mutant cells is not yet conclusive. Our working model is that activation of the PI3K-Akt-mTORC1 pathway drives a metabolic rewiring of the cell, redirecting glycolytic metabolism and promoting lactate production. Although the underlying mechanisms need further investigations, inhibition of the PI3K-Akt-mTORC1 axis reduces mutant load and improves bioenergetic competence in a cell-autonomous way. Thus, we propose that nutrient-sensing pathways that may have evolved as adaptive responses to altered cellular metabolism prove to be maladaptive, driving a positive feedback cycle that may shape the phenotype of the disease and determine disease progression. These data strongly suggest that cell signalling pathways activated by altered metabolism represent potential therapeutic targets that may benefit people suffering from diseases caused by mtDNA mutations.

## Methods

**Cell lines**. The A549 cybrid cell line derived from the fusion of enucleated patient cells harbouring the m.3243 A > G mtDNA mutation was a gift from Ian Holt (Biodondostia Research Institute, San Sebastián, Spain). The 143B cybrid cell lines with ~50% and ~80% of the m.8993 T > G mtDNA mutant loads were gifts from the Minczuk lab (MRC Mitochondrial Biology Unit, Cambridge, UK). Control and patient fibroblasts bearing the m.3243 A > G mtDNA mutation were obtained from the MRC Centre for Neuromuscular Disorders Biobank London. The patient fibroblasts with the m.3243 A > G mutation were isolated from two female subjects (mother and daughter) at the age of 59 (patient 1) and 35 (patient 2) at the time biopsies were taken. For patient 1, the clinical symptoms included diabetes, myoclonus, sensorineural hearing loss, memory decline, myopathy, pigmentary retinopathy and bipolar affective disorder. For patient 2, the symptoms included diabetes, sensorineural deafness, cerebellar ataxia, myopathy, epilepsy, depression and cognitive impairment. These cell lines were cultured in Dulbecco's modified Eagle's medium (Gibco #10566016) supplemented with 10% foetal bovine serum (Gibco #16140071), and 1% Antibiotic-Antimycotic (Gibco #15240096) and incubated at 37 °C with 5% $CO_2$.

**Human muscle biopsies**. The muscle biopsies were from a female patient with the m.3243 A > G mutation (at the age of 55) and a matched healthy control. The study was approved by the Queen Square Research Ethics Committee, London (09/H0716/76). Informed consent was obtained from all participants.

**Cell culture and drug treatments for PI3K, Akt or mTORC1 inhibition**. A549 cybrid cells (WT and mutant) and 143B cybrid cells were passaged every 3–4 days at 80% confluence and trypsinised using 0.25% trypsin-EDTA (Gibco #25200056). Similarly, control and patient fibroblasts were passaged every week in the same way. Media with or without drugs were changed every 2–3 days. For drug treatments, rapamycin (Cayman Chemical #13346), LY294002 (Cayman Chemical #70920) GDC0941 (Cayman Chemical #11600) and MK2206 (Cayman Chemical #11593) were dissolved in DMSO (Sigma-Aldrich #D2650) at 10 mM stock

concentrations. Drugs were subsequently diluted to their working concentrations in media. Heteroplasmy level in the A549 cybrid cells, the 143B cybrid cells and the fibroblasts of patient 1 without treatments did not change significantly throughout the study as assessed by ARMS-qPCR or PCR-RFLP.

**Measurements for mtDNA mutations**. Levels of mtDNA mutation were detected using PCR-restriction fragment length polymorphism (RFLP)[9,47] or allele refractory mutation system (ARMS)-based quantitative PCR (qPCR) analysis[10]. DNA extractions from frozen cells were performed using the DNeasy Blood & Tissue Kit (Qiagen #69506). Concentrations of DNA samples were quantified using Nano-Drop. All primer pairs used can be found in Table S5. For PCR-RFLP, samples (20 ng/µl, 2 µl) were mixed with the master mixes containing 1 µl primers (10 µM each) and reagents of GoTaq G2 (Promega #M7845) and the final volume was 25 µl. After 30 thermocycles, amplified PCR products (8 µl) were further digested by ApaI (Promega #R6361) for the m.3243 A > G and HapII (Promega #R6311) for the m.8993 T > G and analysed by a 2% agarose gel with ethidium bromide. Densitometric analysis was performed by Fiji[48]. For ARMS-qPCR, samples were diluted to 0.4 ng/µl. ARMS primer working solutions (5 µM, 1 µl each) and SYBR Green JumpStart Taq ReadyMix (Sigma-Aldrich #S4438) were added together as master mixes for mutant and wild-type genes. DNA samples (3 µl) and master mixes (7 µl) were pipetted into a 96-well PCR plate (Bio-Rad #MLL9651) and PCR amplification was performed using the CFX96 Touch Real-Time PCR Detection System (Bio-Rad). Each sample has three technical replicates. Mutant heteroplasmy level (%) was calculated using $1/[1 + (1/2)^{\Delta CT}] \times 100\%$, where $\Delta C_T$ (cycle threshold) $= CT_{wildtype} − CT_{mutant}$.

ARMS-qPCR is a quantitative way to measure mutant load. In contrast, PCR-RFLP is relatively a qualitative way to detect mtDNA mutations. Due to an intrinsic problem of PCR-RFLP, part of the PCR products are hybrids which have a strand of wildtype (WT) and a strand of mutant, because a single nucleotide mutation is not strong enough to avoid their annealing. Moreover, restriction enzymes cannot recognise and cleavage these hybrids of WT/mutant double-strand DNA, which means that in the 'WT' band, part of the PCR products are hybrids. Thus, the PCR-RFLP largely underestimates the mtDNA mutant loads, which is the reason why there are weaker bands of the mutant than of the WT.

**Quantification of relative mtDNA copy number**. The relative mtDNA copy number of cells was determined using quantitative PCR with primers for the mtDNA tRNA[Leu(UUR)] and with primers for the nuclear B2-microglobulin as previously described[49]. Standard 96-well PCR plate with optically clear sealing film and CFX96 Real-Time PCR Detection System (Bio-Rad) were used. PCR mix consists of 2 µl of template DNA (3 ng/µl), 2 µl of primer pair (final concentration of 400 nM), 12.5 µl of SYBR Green JumpStart Taq ReadyMix (Sigma-Aldrich) and 8.5 µl of DNase/RNase free $H_2O$. The thermal cycling conditions were as follows: 50 °C for 2 min, 95 °C for 10 min and then 40 cycles of 95 °C for 15 s and 62 °C for 1 min. Each sample has three technical replicates. The following equation was used to determine the relative mitochondrial DNA content, $2 \times 2^{\Delta CT}$, where $\Delta C_T$ is nuclear DNA CT value subtracted by mtDNA CT value.

**The mitochondrial oxygen consumption rate**. Measurements of aerobic respiration and glycolysis were conducted with the Seahorse Bioscience XFe96 bioanalyzer using the Seahorse XF Cell Mito Stress Test Kit (Agilent #103015-100). Cells were seeded on XF96 cell culture microplates (Agilent #102416-100) 2 days before the experiment (A549 cybrid cells, $1 \times 10^4$ cells/well; Fibroblasts, $2 \times 10^4$ cells/well). On the day of the experiment, the culture medium was replaced with Seahorse XF Base medium (Agilent #103334-100) supplemented with 1 mM pyruvate (Gibco #11360070), 2 mM glutamine (Gibco #25030081) and 10 mM glucose (Gibco #A2494001) and incubated for 30 min at 37 °C in a $CO_2$-free incubator before loading into the Seahorse Analyser. After measuring basal respiration, the drugs oligomycin (5 µM), FCCP (1 µM, 2 µM), and rotenone/antimycin A (0.5 µM/0.5 µM) were added to each well in sequential order. Data were analysed using the XF Cell Mito Stress Test Report Generator. After the assay, cells were stained with Hoechst 33342 (5 µM; Thermo Scientific #62249) for 30 min. ImageXpress was then used to count the numbers of cell nuclei (cell numbers) in each well. The normalisation of the experiments is based on the relative cell numbers obtained.

**Mitochondrial membrane potential (Δψ$_m$)**. Cells were seeded ($3 \times 10^4$/well for A549 cybrid cells; $1 \times 10^4$/well for fibroblasts) in glass-bottom 24-well plates, 2–3 days before imaging. Cells were washed twice with the recording medium, which was phenol red-free DMEM (Gibco #A1443001) with 10 mM glucose, 1 mM glutamine, 10 mM HEPES, adjusted to pH 7.4; and then incubated with 25 nM tetramethylrhodamine methyl ester (TMRM) for 30 min at 37 °C. Cells were imaged with an LSM 880 (Carl Zeiss) confocal microscope using Fluar 63x/1.40 oil immersion objective lens at 37 °C. TMRM was excited with a 561 nm Argon laser with an output power of 0.2 mW. MBS 488/561 was used as a beam splitter and emitted fluorescence collected at 564–740 nm. Images were acquired using Zen Black software (Carl Zeiss) and fluorescence intensity was quantified using Fiji with the same threshold across all samples.

**ROS measurements**. The rate of general ROS production was assessed using dihydroethidium (DHE, Invitrogen #D11347), an intracellular superoxide indicator which is oxidised by superoxide to ethidium which fluoresces red. As for mitochondrial ROS production, MitoSOX (Invitrogen #M36008) was used. One day before the experiment, cells were seeded in a glass-bottom 96-well plate (SensoPlate #655892) at a density of $2 \times 10^4$ cells per well. On the day of the experiment, cells were washed twice with PBS and incubated with 5 μM DHE or MitoSOX in a recording medium (phenol red-free DMEM, 10 mM glucose, 1 mM glutamine, 10 mM HEPES, adjusted to pH 7.4). Measurements of fluorescence intensity were taken at intervals of 5 min at 37 °C using the CLARIOstar microplate reader (excitation/emission = 518/606 nm for DHE excitation/emission = 510/580 nm for MitoSOX). The total incubation time for DHE was 1 h and 30 min for MitoSOX. The slope of fluorescence intensity was calculated (Supplementary Fig. 1e) and represents the rate of ROS production.

**Glucose uptake using 2-NBDG**. The fluorescent deoxyglucose analogue, 2-NBDG (Invitrogen #N13195), was employed as a probe to measure rates of glucose uptake by cultured cells. Cells were seeded in a glass-bottom 96-well plate as described for ROS measurements. On the day of experiments, cells were firstly washed with glucose-free recording media twice and incubated with the glucose-free media for 1 h. After media aspiration, recording media with 2-NBDG (100 μg/ml) were then added into wells and incubated for 30 min. After the incubation, cells were then again washed with glucose-free recording media twice and fluorescence intensity was measured using the CLARIOstar microplate reader (excitation/emission = 467/542 nm).

**Medium pH values**. Medium pH values were measured based on the ratiometric property of phenol red, a common pH indicator in media. The absorbance of phenol red changes in response to changing pH. Cells were seeded in 96-well plates at a density of $1 \times 10^4$ cells per well with 200 μl media and cultured for 2 days. On the day of experiments, media of each well were then transferred to a new 96-well plate and the absorbance of phenol red at 443 and 570 nm was measured immediately. The higher the absorbance ratios of 443 to 570 nm, the more acidic the media.

**Glucose and lactate concentrations in media using CuBiAn**. Media for pH measurements were then directly used for quantifying glucose and lactate concentrations in media. Samples and fresh media were measured using the CuBiAn HT-270 biochemistry analyser (Optocell technology) with its Glucose (#200106) and Lactate Assay Kits (#200115) according to the manufacturer instructions.

**Blue native gel electrophoresis (BNGE) and in-gel activity assays**. Mitochondria were isolated from cultured fibroblasts and cell lines according to the method described earlier (ref). Digitonin-solubilized mitochondria proteins (100 μg) were separated on precast 3–12% gradient blue native gels (Invitrogen #BN1001) according to the manufacturer's instructions. After electrophoresis, the gels were electroblotted onto PVDF membrane (Millipore #IPVH00010) and probed with anti-OxPhos antibody cocktail (1:1000, Invitrogen #45-8199), anti-SDHA (1:1000, Abcam #ab137040) and anti-ATP5A (1:1000, Abcam #ab14748). The enzymatic activity of different OxPhos complexes is determined by in-gel assays. For CIV + CI activity, the gels were incubated first in CIV substrate (50 mg diaminobenzidine and 100 mg cytochrome c in 50 mM phosphate buffer, pH 7.4) until a brown signal was observed and then incubated in CI substrate (0.1 mg/ml NADH, 2.5 mg/ml Nitrotetrazolium Blue chloride in 100 mM Tris–HCl, pH 7.4) until blue signal appeared. The reaction was stopped with 10% acetic acid and the gels were washed and scanned.

**SDS-PAGE and immunoblotting**. For immunoblotting, cells were seeded 1-day prior experiments (60 mm plates for the A549 cybrid cells and 10 cm plates for fibroblasts) and then washed with ice-cold PBS once and lysed using 150–300 μl RIPA buffer (Sigma-Aldrich #R0278) with one cOmplete™ Protease Inhibitor Cocktail (Roche #4693116001) tablet and one PhosSTOP Phosphatase Inhibitor Cocktail (Roche #4906837001) tablet. Cells were then scraped and centrifuged at 16,000xg at 4 °C for 30 min. Protein concentration in the supernatant was quantified using the Pierce BCA Assay Kit (Thermo Scientific #23227). For immunoblotting, 30 μg of protein samples in NuPAGE 4x LDS Sample Buffer (Invitrogen #NP0007) and 2% β-mercaptoethanol (Sigma-Aldrich #63689) were boiled at 99 °C for 5 min. Proteins were separated on 4–12% NuPAGE Bis-Tris polyacrylamide gels (Invitrogen #NP0335) or 12% Bis-Tris gels (#NW00122) immersed in MOPS running buffer (Invitrogen #NP0001) and transferred onto PVDF membranes (Millipore #IPFL00010). Membranes were then incubated in Intercept (TBS) Blocking Buffer (Li-COR Biosciences #927-60001) for 1 h at room temperature. After the addition of primary antibodies diluted in the blocking buffer with 0.1% Tween-20, membranes were incubated overnight at 4 °C on a shaker. Multiple primary antibodies were incubated with membranes at the same time (please see supplementary data of immunoblotting raw images for details). Subsequently, membranes were incubated with appropriate secondary antibodies (Li-COR Biosciences; 1:10000; IRDye® 680RD Goat anti-Mouse IgG, #926-68070; IRDye® 800CW Goat anti-Rabbit IgG, #926-32211) for 1 h at room temperature before

signals were developed with the Li-COR Odyssey CLx system. For protein targets for which we have measured both phosphorylated and total forms, the phosphorylation form of the protein was probed first, followed by the total form. Details of all the antibodies used in this study can be found in Supplementary Table 4. Following are the antibodies and dilutions for immunoblotting: Anti-PC (1:1000, Novus Biologicals #NBP1-49536), anti-phospho-PDHA (Ser293, 1:1000, Millipore #AP1062), anti-PDHA (1:1000, Invitrogen #45-6600), anti-p-Akt (Ser473, 1:1000, Cell Signaling Technology #9271), anti-Akt (1:3000, Cell Signaling Technology #9272), anti-p-mTOR (Ser2448, 1:1000, Cell Signaling Technology #5536), anti-mTOR (1:3000, Cell Signaling Technology #2983), anti-p-S6 (Ser235/236, 1:3000, Cell Signaling Technology #4858), anti-S6 (1:3000, Cell Signaling Technology #2217), anti-p-AMPKα (Thr172, 1:1000, Cell Signaling Technology #2535), anti-AMPKα (1:3000, Cell Signaling Technology #2532) and anti-β-actin (1:10000, Santa Cruz Biotechnology #sc-47778).

**Live-cell imaging for cell growth and death using the Incucyte platform**. Cells were seeded at a density of 2000 cells/well in a 96-well plate 1 day prior to imaging with regular cell culture media. For cell growth/death in a variety of nutrient conditions, cells were washed with PBS twice and then DMEM (Gibco #11966025 or #11960044) with different concentrations of glucose, galactose (Sigma-Aldrich #G5388) or glutamine and 10% dialysed FBS (Sigma-Aldrich #F0392) were added. For drug treatments, media were then replaced with fresh media with/without drugs along with 20 nM SYTOX Green Nucleic Acid Stain (Invitrogen #S7020) just before imaging. Images were acquired using the Incucyte with 20X objective every 2 h for a 3–4 day period. Cell confluency and cell death numbers were analysis on Incucyte software and sequentially extracted. Rates of cell growth were analysed by fitting the growth curve to the exponential cell growth model (Supplementary Fig. 2a) using the solver in Excel, while cell death was quantified by simply normalising the numbers of dead cells (SYTOX Green stained) to total cell confluency.

**Measurements of cytosolic NADH:NAD⁺ using SoNar**. The genetically encoded probes, SoNar for NADH:NAD⁺ ratio, were developed by and obtained from the Yang's Lab (Chinese Academy of Sciences)[11]. For A549 cybrid cells, the cells were seeded at the density of $4 \times 10^4$/well in glass-bottom 24-well plates 2 days before transfection. Lipofectamine 3000 (Invitrogen #L3000001) was used for transfection according to the manufacturer's protocol. Specifically, we used 1 μg DNA and the ratio of p3000 to the 3000 reagent was 2:1. The cells were imaged 2 days after transfection in recording media with the LSM 880 microscope using a 20x/0.8 objective lens at 37 °C. Both probes were excited at 405 and 488 separately and emitted fluorescence longer than 535 nm was collected. For fibroblasts, Human Dermal Fibroblasts Nucleofector Kit (Lonza #VPD-1001) was used for transfection according to the manufacturer's protocol. Specifically, we used 2.5 μg of DNA for $5 \times 10^5$ cells. Similarly, fibroblasts were imaged 2 days after transfection with the same conditions as A549 cybrid cells. Images were acquired using Zen Black software (Carl Zeiss) and analysed using Fiji. Regions of interest (ROI) were generated by thresholding and binarizing images and average fluorescence intensity in the ROIs for each channel were then measured. Ratios between the signal excited at 405 and 488 nm were calculated.

**Metabolomics**. Control and patient fibroblasts were plated 24 h before the experiment in 10 cm dishes ($1 \times 10^6$ cells/dish). On the day of experiments, the medium was replaced with phenol red-free DMEM with 1 mM glutamine, 10% dialysed FBS (prepared as described in Yuneva et al., 2007) and 10 mM [U-¹³C]-glucose (Goss Scientific #CLM-1396-1). After 24 h incubation, cells were washed once with ice-cold PBS. Metabolites were extracted by adding 500 μl of ice-cold methanol, scraping, and transferring to tubes on ice. Dishes were then subsequently washed with 250 μl methanol and 250 μl water containing 2.5 nM of each nor-leucine and scyllo-inositol per sample as internal standards. Finally, 250 μl of chloroform was added and samples were sonicated for three rounds (8 min each) at 4 °C. After centrifugation (18,000xg for 10 min at 4 °C), the upper polar fraction was collected and used for GC–MS analysis, while the protein pellets were dried, lysed using 62.5 mM Tris buffer (pH 6.8) containing 2% SDS and used for protein quantification using BCA assay (Abcam). Briefly, the polar fraction was washed twice with methanol, derivatized by methoximation (Sigma, 20 μl, 20 mg/ml in pyridine) and trimethylsilylation (20 μl of N,O-bis(trimethylsilyl) trifluoroacetamide reagent (BSTFA) containing 1% trimethylchlorosilane (TMCS), Supelco), and analysed on an Agilent 7890A-5975C GC–MS system. Splitless injection (injection temperature 270 °C) onto a 30 m + 10 m × 0.25 mm DB-5MS + DG column (Agilent J&W) was used, using helium as the carrier gas, in electron ionisation (EI) mode. The initial oven temperature was 70 °C (2 min), followed by temperature gradients to 295 °C at 12.5 °C/min and then to 320 °C at 25 °C/min (held for 3 min). Data analysis and peak quantifications were performed using MassHunter Quantitative Analysis software (B.06.00 SP01, Agilent Technologies). The level of labelling of individual metabolites was corrected for the natural abundance of isotopes in both the metabolite and the derivatization reagent. Abundance was calculated by comparison to responses of known amounts of authentic standards[50,51]. The data were further processed and visualised by MetaboAnalyst 5.0, using the Pathway Analysis module (data were normalised to

controls and scaled by the standard deviation of each variable) with the library of human SMPDB[52].

**RNA-sequencing.** Fibroblasts for RNA-sequencing were plated in 10 cm dishes with regular media ($1 \times 10^6$ cells/dish) 1 day before RNA extraction. RNA extractions were performed using the RNeasy Mini Kit (Qiagen #74104). Concentrations and quality of RNA samples were quantified using NanoDrop (total RNA >250 ng, $A_{260}/A_{280} = 1.8$–2.1, $A_{260}/A_{230} > 1.7$). RNA-sequencing was then performed at UCL Genomics and analysed by the SARTools R package[53]. The resulting datasets were further analysed by ingenuity pathway analysis (IPA, Qiagen) with a threshold of FDR <0.05 and visualised by Morpheus (https:// software.broadinstitute.org/morpheus) for heatmaps and NetworkAnalyst 3.0[54] for volcano plots and principal component analysis.

**Immunohistochemistry.** For immunohistochemistry, muscle biopsies were fixed in phosphate-buffered saline (PBS) and fixed in 4% paraformaldehyde (PFA), made in PBS, for 4–8 h at room temperature (RT). Following fixation, tissues were cryoprotected in 30% sucrose in Diethyl Pyrocarbonate (DEPC) treated PBS, embedded and frozen in a mixture of 15% sucrose/50% Tissue-Tek OCT (Sakura Finetek), and sectioned in the coronal plane at 20 μm using a Cryostat (Bright Instruments). Patient muscle biopsies sections were washed in PBS, blocked in a solution of 5% normal goat serum (Merck KGaA) (v/v) containing 0.1% Triton X-100 (v/v) (Merck KGaA) in PBS at RT for 2 h. They were first incubated in primary antibodies at RT overnight. The following antibodies were used: pan-AKT (1:200, Cell Signaling Technology #2920), phospho-AKT (p-AKT; 1:200, Abcam #ab81283), S6 (1:200) and phospho-S6 (1:200). Following incubation in primary antibodies, sections were washed in PBS, incubated in biotinylated anti-species secondary antibodies (1:250; Vector Laboratories) for 2 h. Sections were washed and incubated with bisbenzimide (10 min in 2.5 μg/ml solution in PBS; Merck KGaA). Images were collected using an SP2 Leica confocal microscope (Leica Microsystems, UK). Sequential images were subsequently reconstructed using Metamorph imaging software (Universal Imaging Corporation, West Chester, PA).

**Immunofluorescence.** Fibroblasts were seeded at a density of $5 \times 10^4$ cells/well in 24-well plates on 10 mm coverslips. After 24 h incubation, cells were treated with LY or RP for 4 weeks and 8 weeks, respectively. After the long-term treatment, the cells were washed thrice with 1X PBS and fixed in 4% paraformaldehyde for 15 min at room temperature and permeabilised with 0.1% Triton X-100 for 15 min in PBS. The cells were then washed and incubated with an anti-MT-COI antibody (1:100, Abcam #ab14705) in 3% BSA for 1 h at room temperature followed by incubation with Alexa Fluor 488-conjugated secondary antibody (Invitrogen #A11029) for 1 h at the room temperature. After antibody labelling, the coverslips were mounted on a glass slide using ProLong™ Gold Antifade mountant with DAPI and imaged using the confocal microscope as described above.

**TaqMan SNP genotyping for single-cell qPCR.** Primers for distinguishing WT and the m.3243 A > G mutant mtDNA were designed by submitting the region of the m.3243 A > G mutation (~120 bp) to the website of online Custom TaqMan® Assay design tool (https://www.thermofisher.com/order/custom-genomic-products/tools/genotyping/) The Custom TaqMan SNP Genotyping Kit (Applied Biosystems #4332073) was ordered and prepared according to the manufacturer instructions. For single-cell sorting, cells were resuspended ($1 \times 10^6$ cells/ml for A549 cybrid cells; $5 \times 10^5$ cells/ml for fibroblasts) in 1 ml PBS with 1% FBS, filtered through the 70-μm mesh and then kept on ice. Sorting cells to 96-well PCR plates using BD FACS Aria Fusion with BD FACSdiva 7 was performed by Flow Cytometry Core Facility, Division of Medicine at UCL. Cells in 96-well plates were lysed with 2 μl of 0.2% Triton X-100 and kept at −80 °C immediately. For qPCR, TaqPath ProAmp Master Mix (Applied Biosystems #A30866) was mixed with primers of TaqMan SNP genotyping and added into 96-well PCR plates to reach a final volume of 10 μl in each well. Thermocycles were set according to the manufacturer's instructions. Mutant loads were determined by the ratio of mutant to total fluorescent intensity.

**Quantification and statistical analysis.** All statistical analyses, unless otherwise stated in figure legends, were carried out using GraphPad Prism 8. To compare the means between two groups, a two-tailed unpaired $t$-test was used for normally distributed data. One/two-way ANOVA with Tukey's multiple comparisons test was performed for multigroup (at least three) comparisons. Data were presented as graphs displaying mean ± SD, of at least three independent biological replicates. Means of control samples on immunoblotting or immunofluorescence are typically centred at one (or 100%) to ensure easier comparisons unless otherwise noted. Differences were only considered to be statistically significant when the $p$ value was less than 0.05. Estimated $p$ values are either stated as the actual values or denoted by *$p < 0.05$, **$p < 0.01$, ***$p < 0.001$, ****$p < 0.0001$.

For detailed processing and statistical analysis of RNA-sequencing and metabolomics, please find them in the previous section with the same subheadings.

No statistical method was used to predetermine sample size, and replicates are shown in Figure legends. The investigators were not blinded to allocation during experiments and outcome assessment.

**Reporting Summary.** Further information on research design is available in the Nature Research Reporting Summary linked to this article.

## Data availability

Raw and processed RNA-seq data are accessible at the Gene Expression Omnibus under accession GSE175477. Other data including uncropped images of immunoblotting are available in the Source Data files of this paper. Source data are provided with this paper.

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

## Acknowledgements

We thank Alice Giustacchini (UCL ICH) for help with single-cell analyses, Stephanie Carrington (Division of Neuropathology, UCL IoN) for sectioning the muscle biopsies, Ash Merve (Consultant Neuropathologist, Division of Neuropathology, UCL IoN and Department of Histopathology, Camelia Botnar Laboratory, Great Ormond Street Hospital) for reviewing the biopsy and taking H&E stained images, and Metabolomics STP of the Francis Crick Institute for assistance with GC–MS analysis. We also acknowledge The MRC Centre for Neuromuscular Diseases Biobank (supported by the National Institute for Health Research Biomedical Research Centres at Great Ormond Street Hospital for Children NHS Foundation Trust and at University College London Hospitals NHS Foundation Trust and University College London) for providing all patient-derived cells used in this study. R.D.S.P. is supported by a Medical Research Council (UK) Clinician Scientist Fellowship (MR/S002065/1) and a Medical Research Council (UK) strategic award to establish an International Centre for Genomic Medicine in Neuromuscular Diseases (ICGNMD) (MR/S005021/1). G.S. is supported by Telethon GGP16026 and CRUK Pioneer Award C28472/A29264. M.Y. is supported by the Francis Crick Institute which receives its core funding from Cancer Research UK (FC001223), the UK Medical Research Council (FC001223), and the Wellcome Trust (FC001223). G.E.V. is supported by the National Agency for Research and Development (ANID)/ Scholarship Programme/DOCTORADO BECAS CHILE/2019 – 7220052 We also thank Action Medical Research for funding the early stages of this work and the Ministry of Education, Taiwan for funding C.-Y.C.'s Ph.D. by the Government Scholarship to Study Abroad.

## Author contributions

C.-Y.C. drove most of the project, conducted or supervised most of the experiments himself and wrote the first draft of the manuscript. K.S. carried out Blue Native page gels and in-gel activity assays and analysed these data. V.N.K. carried out crucial pilot experiments as proof of principle and first identified the activation of PI3K-AKT and mTORC1 in the mutant cells. G.E.V. performed immunoblotting, especially following drug treatments. J.H.A., E.T. and J.T. carried out replicates of bioenergetic analyses especially helping to establish the reduction in mutant load and improved bioenergetics in response to long-term inhibitors of PI3K, Akt or mTORC1. W.D.A. performed immunofluorescence labelling of patient muscle biopsies. B.B. carried out some of the immunoblotting relating to PI3K-AKT-mTORC1 phosphorylation and gave considerable guidance and advice on this aspect of the work. R.D.S.P. was instrumental in obtaining patient-derived cells and muscle biopsies from patients. G.S. was deeply involved in discussions and made invaluable suggestions especially early in the work. M.Y. directed the metabolomic experimental protocols and supervised the analysis and interpretation of those data. K.S., G.E.V., W.D.A., G.S. and M.Y. edited the draft of the paper. M.R.D. conceived and directed the work and wrote the final draft of the paper.

## Competing interests

The authors declare no competing interests.
