## [Peer Review File · Nature Communications]

Constitutive activation of the PI3K-Akt-mTORC1 pathway sustains the m.3243A>G mtDNA mutationREVIEWER COMMENTS

Reviewer #1 (Remarks to the Author):

The manuscript is submitted by the Duchen lab, which has a solid history of key discoveries in mitochondrial biology. In this manuscript, the authors propose chronic P13K-Akt-mTORC1 activation in the context of m.3243A>G disease pathology. Coupled with this, are suggested phenotypes of redox imbalance, glucose dependence and effects on autophagy.

This manuscript is, in general, well written. However, the density of panels (I lost count) and the sheer range of phenomenology in each figure panel obscures the clarity of the narrative. Thus, it is easy for even the expert reader to become completely lost. The reviewer recognises that the authors have invested a lot of time and conducted a lot of work here. However, - the density of data presents a challenge to the reader, and it was a struggle to get through. In its current formulation, it is an impenetrable manuscript.

Notwithstanding the above, this reviewer is generally optimistic about the findings in this paper, and their novelty for the mitochondrial disease field and broader cell biology community. Some of the conclusions of the authors are acceptable if they can address the identified caveats through targeted refinement. Far more clarity needs to be introduced in the take-home message. The clearest and most reliable result is the chronic Akt activation in a model of mitochondrial disease, and there are tantalising human data to suggest this could be a clinically-relevant feature in vivo. However, the accuracy and precision of the signalling claims require additional experimental verification in light of the limitations posed by the reagents used. With the proposed essential revisions and a revised interpretation, the manuscript would be acceptable for eventual publication. Significant concerns are detailed on a figure-by-figure basis below and should be addressed comprehensively by the authors. The authors would probably make the manuscript shine more by leading with less.

General technical comment:

The authors are strongly encouraged to review the raw data with a specific focus on instances where statistical significance is claimed in the presence of pronounced variability. For example, the error bars overlap in Figure S1 Jii, and the fold change is <1. How can there be a significant result if the variability is so large between experimental samples? This result and others like it seem unconvincing and detract from the paper. The authors are encouraged to be particularly forensic concerning the inclusion of such data.

Additionally, the formatting of graphs is quite chaotic. Please make all diagrams consistent with one another (preferably all scatter plots) - it was hard work to follow this paper.

General overall comment:

The title of the paper does not reflect the content - and is confusing to the reader. Please use precise phrasing. Pending the experimental revisions, I suggest: "Chronic activation of the Akt pathway defines the m.3243A>G mtDNA mutation."

The most convincing aspects of the paper are the Akt, metabolic and mitochondrial biology results. The data on autophagy and lipid droplets is less compelling and looks very out of place compared to the rest of the manuscript. I understand the drive to have an overarching narrative, but the amount of work to substantiate the claims of mitophagy would render this paper unsuitable for publication. The authors should consider omitting the autophagy/mitophagy/lipid droplet data and focusing on the most specific, most reproducible phenotypes - in this case, chronic Akt activation in vitro and in vivo.

Detailed Comments

Figure 1 and 1S

A significant concern for the entire study is that uridine is not a constituent of the cell culture media. It is well established that a defective respiratory chain impairs cell proliferation. Thus, uridine supplementation in the media is a widely used additive in the culture of mtDNA mutant/patient cell lines. As shown in Figure S1 F1, the growth rate of the patient cell line is lower than the controls. Please clarify the reasons for culturing in this unconventional manner. Also, please see the following landmark papers in the field to support this claim: doi: <https://doi.org/10.1016/j.celrep.2020.107538>

doi: [10.1073/pnas.1906896116](https://doi.org/10.1073/pnas.1906896116)

doi:10.1126/science.2814477 - do these phenotypes exist in the presence of uridine? What is the Akt status like in these cells \pm uridine?

In Figure 1S, the authors show mtDNA copy numbers for A549 cybrid cell lines. In figure S7A, they follow the time-dependent mtDNA copy number change in different conditions. However, data for mtDNA copy number of the patient cell lines compared to control cell lines is missing.

Figure 1S shows the membrane potential in A549 cells, but it does not show the morphology data as shown for the patient cell lines in Figure 1F.

Since the metabolic phenotype presented in Figure S1 does not match or comparable to what they have shown for the patient cells in Figure 1, the use of A549 cells to study the mechanism of 3243A>G cells is unconvincing.

The way authors describe patient cells growth rate and mutant cells growth rate (lines 110-112) is confusing for readers. Please state clearly the cell line name as 3243A>G cells or A549 cell line. It should be effortless for the reader to know what they are studying.

The NADH: NAD result is very curious indeed. As referenced above, how can there be statistical significance with such experimental variability? This does not compute with such error bars. Please review these data.

Figure 2 and 2S

The data presentation is meandering. Starting with supplemental figures, returning to the central figure and back again - it becomes irritating and easy to lose interest. The authors are encouraged to think deeply about figure presentation and depict the results in the most straightforward fashion possible.

Much of the analysis is dependent on metaboanalyst, which, although convenient, can vary wildly depending on the selected input parameters. The authors should model the data directly to check if these outputs reflect the best model possible.

The metabolomics data from the control samples looks highly variable; why is this? Please address directly.

The authors show two types of patient cells in Figure 1. Which line is used for metabolomics and flux?

It is not surprising that lipid droplets form upon respiratory stress. All mitochondrial stressors promote the formation of LDs to some extent. The images are very unusual and look overexposed. Please use a conventional BODIPY dye or omit. I do not see what this data adds to the paper.

Figure 3

This is a very disjointed and confusing figure. Nowhere in the RNA-seq data is autophagy implicated. Everything appears to be very convoluted. Even ULK1 - a master kinase involved in autophagy seems unaltered between the controls and patient fibroblasts in the supplemental data.

Same question as for Figure 2. Which patient cells were used for RNAseq?

"These results suggest that the constitutive induction of PI3K-Akt-mTORC1 signalling remodels cell metabolism" is a central thesis of the manuscript. All the raw blotting data should be presented to demonstrate this.

In general, the quality of biochemistry could be much better, and as it stands, it is mechanistically too simplistic. The authors should include full-length blots in supplemental data.

The Akt data are very exciting (the most novel part of the paper) but need to be clarified, is this an activating or inhibitory PTM? "Phosphorylation" can mean anything. Is it Thr309 or Ser473? If it is activating, the authors should include monitoring the Akt-dependent phosphorylation of a well-known substrate (e.g. T246 on PRAS40, T346 on NDRG1 or Ser939/Thr1462 of TSC2) to solidify this.

Is there a downregulation of a particular phosphatase in the mtDNA patient cell line that could account for this chronic Akt activation? e.g. PP2A (DOI: 10.1126/science.1106148) or PHLPP (DOI: 10.1016/j.molcel.2005.03.008)

There is no clear rationale for investigating autophagy here, although it is not surprising that it is altered in mitochondrial disease. But this adds nothing to the story. The LC3 immunoblots are not convincing. The levels of LC3 are known to fluctuate between individuals, and thus, inhibitors must be used. Bafilomycin A1 should be used, and all samples should be resolved on the same 13% gel \pm starvation.

As an extension, the quantitation of the blots is problematic. Avoid quantitation by ImageJ/FIJI for western blotting - please use the values provided by the dedicated LiCOR CLx software using total protein normalisation.

The ISR data is not convincing and confuses the manuscript once again. ATF4 and particularly CHOP antibodies, whilst widely bought and sold - are notoriously non-specific. Hence the reason for the use of CHOP: XFP reporters by pioneering UPR labs. What does the RNA-seq data say about the ISR - this would be more convincing.

The term "metabolic stress" is vague. Which metabolic stress? It is a respiratory deficiency.

The use of the mCherry-GFP-LC3 reporter is more convincing than the biochemical data. However, 20 cells are not sufficient to draw any conclusion. FACS based quantitation of the mCherry-only/total GFP ratio enables quantitation of >10-50K cells. This should be employed \pm BafA1 (much more preferable to CQ which is not used much in the field anymore), or high content microscopy analyses (>50-150 cells, dozens of fields) of the reporter should be added to strengthen the data. However, this data is extremely confusing in the manuscript and should really be omitted.

The muscle biopsy data is excellent! The authors should consider including staining for an Akt substrate (as above) to verify activation at the substrate level in vivo.

Figure 4

I understand the author's interpretation of the data shown in B, pharmacological inhibition of PI3K-Akt-mTORC1 signalling is in general, affecting the mutant load in A549 cells and patient cells. However, this is a major oversimplification in terms of what we know about the mechanism of Akt signal transduction. The author's interpretation is predicated on the use of LY294002 - widely used but well known non-selective inhibitor of PI3K. LY294002 inhibits TORC1, PLK1, CK1, PIM1/2, HIPK2 and GSK3 in addition to robustly binding non-kinase ATP-binding proteins (DOI: 10.1042/BJ20061489; DOI: 10.1016/j.chembiol.2004.11.009; DOI: 10.1042/BJ20070797). Accordingly, the authors need to revise these experiments and use much more validated inhibitors for Class I PI3Ks.

Accordingly, the authors should perform revised experiments using allosteric Akt inhibitors known to selectively inhibit phosphorylation of the T-loop/other motifs in Akt, e.g. MK-2206 or AZD5363. For targeted mTORC inhibition, AZD8055 should be used. In light of these comments, all of this data should be re-examined, and consistent with this, changes in C are not convincing. In the repeat experiments, the authors should include uncropped, full-length blots in supplemental data. The authors should use minimal image processing for western blots in the revised manuscript.

Figure 5

The data in Figure 5 are complicated by the absence of uridine and presence of non-selective inhibitors. These data should be repeated to verify the mechanism. The death effects could easily skew the interpretation of load.

Figure S5 i and ii treatment weeks do not match.

Figure 6

Pharmacological inhibition clearly affects the m.3243A>G mutant load, but these effects cannot be attributed to the mechanism postulated due to the specificity of the inhibitors used.

The mitophagy data are not convincing and confusing as presented. It is hard to know what to focus on with 30 small panels of microscopy images. It is unclear how mitophagy is distinct from alterations in general macroautophagy. It is likely the effects the authors are observing are a function of altered macroautophagy - furthermore, mitophagy can occur independently of LC3.

The Keima quantitation is very confusing, and the quantitation does not reflect the conclusions. Several thousand cells need to be analysed by FACS for mt-Keima due to the variance caused by spectral overlap unless STED imaging/spectral unmixing is employed.

Figure 7 (mislabelled as S7?)

LC3B and CytC colocalisation does not look convincing from the given representative image. These are brief snapshots of a dynamic process and can not be used to infer anything. It does not reflect mitophagy, as this can be independent of LC3.

E compares data from patients from Fig 6C. But the data is represented and quantified in a totally different manner and impossible to state anything. Indeed, the controls look different from these images. Please modify the data in exactly the same style and format in the same panel, or omit these data completely.

Reviewer #2 (Remarks to the Author):

The paper by Chung et al., reports on studies in fibroblasts and cybrid cells carrying the m.3243A>G mutation. The authors have extensively characterised changes in bioenergetic, metabolomic, lipidomic and RNAseq profiles in heteroplasmic patient-derived cells and revealed that the mutation promotes upregulation of PI3K-Akt-mTORC1. Pharmacological inhibition of PI3K, Akt, or mTORC1 activated mitophagy, reduced mtDNA mutant load and rescued cellular bioenergetics cell-autonomously, suggesting that activation of the PI3K-Akt-mTORC1 pathway represents a potential therapeutic target.

I have a few questions:

1. On Figure 1 the RFLP shows much weaker band for the mutant than for the wild type. How was the 79% heteroplasmy calculated?
2. What was the rationale behind the reduced Glutamine in the medium? In Figure 1 there is no change on growth rate of low Glutamine, but on Figure 1S on 0.1 mM Glutamine there was a significant growth defect. Please, explain.
3. There was an increase in some amino acids, potentially due to increased uptake. Was there any change detected in the amino acid transporters of these amino acids?
4. How can the authors prove that the PI3K-Akt-mTORC1 signalling is the cause of the metabolic remodelling?
5. While some papers suggest that the activation of the ISR is protective and promote longevity, the data in this paper suggest that the activation of ISR in the context of the m.3243A>G mtDNA mutation is not protective. How can the authors be sure that it is not triggered a protective compensatory mechanism (e.g. metabolic rewiring)?
6. In cell culture it is possible that the mutation load is changing due to the predominant loss of cells with high rate of mutant and defective OXPHOS, and better survival of cells with low or no mutation rate. How can the authors be sure that the treatment caused the reduction of mutation rate? This rate may also be a bit dependent on the starting heteroplasmy rate. It would be good to have some comment on this.
7. How can the authors explain the difference of pathways involved in cells carrying the m.8993T>G mutation and the m.3243A>G mutations? Is the defect detected in m.3243A>G more similar to other defects of mitochondrial protein synthesis? Or this would be specific for m.3243A>G only?
8. Is the improvement only linked to reduced heteroplasmy rate? Can it be influenced by a better metabolic remodelling of metabolism?

9. Immunofluorescence staining revealed a significant decrease in MT-COI expression in patient 1 fibroblasts compared to controls. Was this result only in the cells with higher heteroplasmy rate? It has been reported that the m.3243A>G mutation is affecting more complex I than complex IV in muscle biopsies, so it would be understandable if MT-COI would be only reduced in the more severely affected cells.

10. The mtDNA copy number in the mutant cells was very highly increased (see Fig. S1 A), suggesting a compensatory mitochondrial biogenesis with mtDNA replication. Is it possible that the wild type mtDNA molecules replicated faster?

11. Published RNA-seq data set of muscle biopsies from patients with MELAS detected increase in FGF21 (~48 fold) and GDF15 (~9 fold), these are also present in serum of these patients and therefore suggested to be biomarkers of mitochondrial translation defects. In contrast, in the present study no changes in these two genes were detected in the patient fibroblasts. How can the authors explain this? Are fibroblasts a good model of the metabolic situation in muscle?

12. Some of the western blots in Figure 3 are not really showing the significant defect which is highlighted on the quantification graphs. The blots were repeated for the quantifications. Are there any of the blots better to demonstrate the changes?

Reviewer #3 (Remarks to the Author):

The manuscript by Chung et al. describes the cellular consequences of m.3243A>G mtDNA mutation in patient fibroblast and a cybrid line. Comparison of their results to the published literature, as well as to cybrids carrying m.8993T>G mtDNA mutation shows, that each mutation triggers a different set of cellular responses, and highlights the fact, that not all mtDNA mutations are the same, which is in turn important in design and future studies of possible therapies.

The authors utilize an incredible range of techniques to define the cellular consequences of the mutation and to reveal the upregulation of the PI3K-Akt-mTORC1 axis in patient cells. Further, the authors use a pharmacological inhibition of PI3K, Akt or mTORC1 and show a significant improvement of

practically all studied phenotypes. This is clearly a highlight of the study; however, also the most problematic point, as the authors do not compare these rescues in patient cells to control cells and/or control cells treated with the same drugs. Thus, the reader gets an impression that everything is rescued, although there still might be a significant biochemical defect present. What is the effect of the PI3K-Akt-mTORC1 axis inhibition by rapamycin or LY294002 on control and A549 cells? This major issue needs to be rectified. Regardless of the fact that the rescue might be only partial, the presented results are of great importance to the mitochondrial research field.

Other major comments/questions:

- Although the mutation load is different for patient 1 and 2 (86% and 30%, respectively), there were very little differences in the biochemical phenotype between them? What is known about the threshold for this mutation in MELAS patients? Would it be expected that 30% mutation load is enough to cause such a defect? Also, the mutant load in patient fibroblasts and cybrids reported in Fig. S6B is different from the mutant load reported previously throughout the manuscript. I gather the techniques used are different, but it is worrisome that the precision of at least one of these techniques is not good. Please, comment. What is the reasoning for the increase of the mutation load of patient 2 over time?

- Why was the mtDNA increased 4-5 times in the mutant cybrids? Was the mitochondrial content (measured by enzyme activities/protein levels) changed? Why was the OCR normalized to the mtDNA level? Were all the other results normalized to mtDNA level? What was the OXPHOS supercomplex assembly in these cells? Please, explain the rationale and comment to this in the manuscript.

- Fig. 1E & 4I – the levels of Complex II are also affected in the patient cells, however subunits of this complex are encoded by nuclear genome, and in general Complex II is not affected in patients with mitochondrial mutations, rather often used as a loading control for BN-PAGE? Please, explain.

- There are striking differences in the metabolomics, lipidomics and transcriptomics of the two controls used in the study. This is probably a normal situation, as the cells are not isogenic, but it makes it difficult to compare them to the results in the patient cells. The authors should comment on this in the text.

- Fig. 2C, 3G – the representative WB of patient and/or cybrid cells do not really correspond to the quantification, and in other cases immunoblots are not shown. The visualization of the WB is problematic throughout the manuscript, could the authors improve the signal-to-noise ratio, or show them all in colour?

- Line 202 - Specifically, the increased expression of PC was consistent with the immunoblotting data (Fig. 2C) and correlated with altered TCA cycle activity (Fig. S3G) – this is valid only for patient 2, please comment/correct.

- Fig. 3H – muscle biopsy does not look very good, could an H&E stain be provided to determine that no freezing issues are creating this pattern? Also, have the antibodies used here been tested to indicate that the staining is specific?

- It is interesting to see that the inhibition of PI3K-Akt-mTORC1 axis in m.8993T>G mutant cells had no effect on the mutation load; and highlights the importance, as mentioned above, to study individual

mutants and their effect on cellular metabolism. Did you assess the phosphorylation status and total levels of PI3K-Akt-mTORC1 axis proteins and the downstream reporters of ISR (ATF4, ATF5, CHOP) in m.8993T>G mutant cells and compare them to your cybrids?

- Fig. 5B – the cell growth was significantly increased in patient cells treated with rapamycin, however the text states that this is not the case. Please, clarify.

- Fig. S6D – the figure legend talks about COXIV quantification, however the main text and figure mentions mt-COI? Which protein was quantified? Also, in (ii) and (iii) the patient column is identical, and the two graphs should be merged together and statistical analysis should be done comparing treated patient cells and controls (as mentioned above). (iv) and (v), patient curve is identical, again the graphs should be merged into one.

- It has been shown that LC3 might aggregate in an autophagy-independent manner, thus colocalization with mitochondria might not signify an increase in mitophagy, especially when control cells are not shown. Thus, the use of mt-Keima became the golden standard in determination of mitophagy. Here, however, 543nm-mt-Keima is very strong and practically labels all mitochondria in both patient and control cells? Addition of chloroquine should diminish the 543nm-mt-Keima signal, as the fusion of lysosome and autophagosome is inhibited by it, but this is not the case. It is difficult to assess the level of mitophagy based on the picture provided. Please provide better images.

- Although a possible activation of mtUPR is suggested, no data are shown. Was there any activation of this pathway in the transcriptomics analysis? Did the authors look at the protein levels of mtUPR markers?

Minor comments/questions:

- Scale bars are sometimes missing, especially in zoomed images.

- Fig. S1J - the differences between WT and Mutant are statistically different, but the difference is minor. Could this be caused by the fact that 180 cells were counted, where in other experiments only 50-80 cells are counted?

- Fig. 2D a lot of the fluorescence is outside of the cells, how was the cell area defined, the figure should clearly show the cell outline, otherwise it is meaningless.

- glycerol-3-phosphate is sometimes annotated at G3P, αG3P or alphaG3P - please unify

- Line 339 - MK showed similar results in terms of cell growth and death (Fig. S5A). – Should be Fig. S6A

- Line 509 - (A) PCR-RFLP and ARM-qPCR - should be ARMS-qPCR

- Fig. 5 – please explain pre-treatment/treatment in this experiment

REVIEWER COMMENTS

Reviewer #1 (Remarks to the Author):

The manuscript is submitted by the Duchen lab, which has a solid history of key discoveries in mitochondrial biology. In this manuscript, the authors propose chronic P13K-Akt-mTORC1 activation in the context of m.3243A>G disease pathology. Coupled with this, are suggested phenotypes of redox imbalance, glucose dependence and effects on autophagy.

This manuscript is, in general, well written. However, the density of panels (I lost count) and the sheer range of phenomenology in each figure panel obscures the clarity of the narrative. Thus, it is easy for even the expert reader to become completely lost. The reviewer recognises that the authors have invested a lot of time and conducted a lot of work here. However, - the density of data presents a challenge to the reader, and it was a struggle to get through. In its current formulation, it is an impenetrable manuscript.

We thank the reviewer for this very meticulous critique. We have been working hard under almost impossible conditions to do our very best to address these comments and hope that our revised manuscript will represent a satisfactory improvement. We have edited extensively, and we have removed some data that may have been superfluous to try to streamline the narrative. We do feel that the extensive characterisation of phenotypes is important as all variables may prove to be of value in later screening experiments. However, we hope this is less impenetrable and that clarity is improved.

Notwithstanding the above, this reviewer is generally optimistic about the findings in this paper, and their novelty for the mitochondrial disease field and broader cell biology community. Some of the conclusions of the authors are acceptable if they can address the identified caveats through targeted refinement. Far more clarity needs to be introduced in the take-home message. The clearest and most reliable result is the chronic Akt activation in a model of mitochondrial disease, and there are tantalising human data to suggest this could be a clinically-relevant feature in vivo. However, the accuracy and precision of the signalling claims require additional experimental verification in light of the limitations posed by the reagents used.

We have carried out most of the experiments requested by the reviewers and hope that these will serve to validate the model to your satisfaction!

With the proposed essential revisions and a revised interpretation, the manuscript would be acceptable for eventual publication. Significant concerns are detailed on a figure-by-figure basis below and should be addressed comprehensively by the authors. The authors would probably make the manuscript shine more by leading with less.

See comments above. We agree and have tried to comply.

General technical comment:

The authors are strongly encouraged to review the raw data with a specific focus on instances where statistical significance is claimed in the presence of pronounced variability. For example, the error bars overlap in Figure S1 Jii, and the fold change is <1. How can there be a significant result if the variability is so large between experimental samples? This result and others like it seem unconvincing and detract from the paper. The authors are encouraged to be particularly forensic concerning the inclusion of such data.

Additionally, the formatting of graphs is quite chaotic. Please make all diagrams consistent with one another (preferably all scatter plots) - it was hard work to follow this paper.

We have tried to be more unified. We have changed the formatting of many of the graphs to be internally consistent. All statistical data have been extensively checked and, in some cases, additional experiments performed to increase numbers (please see the source data).

General overall comment:

The title of the paper does not reflect the content - and is confusing to the reader. Please use precise phrasing. Pending the experimental revisions, I suggest: "Chronic activation of the Akt pathway defines the m.3243A>G mtDNA mutation."

We have complied and we have changed the title to: Constitutive activation of the Pi3K-Akt-mTORC1 axis sustains the m.3243A>G mtDNA mutation

The most convincing aspects of the paper are the Akt, metabolic and mitochondrial biology results. The data on autophagy and lipid droplets is less compelling and looks very out of place compared to the rest of the manuscript. I understand the drive to have an overarching narrative, but the amount of work to substantiate the claims of mitophagy would render this paper unsuitable for publication. The authors should consider omitting the autophagy/mitophagy/lipid droplet data and focusing on the most specific, most reproducible phenotypes - in this case, chronic Akt activation in vitro and in vivo.

We are a little troubled by the request to omit the mitophagy data – in our view this provides a mechanism for the changes we see and we felt that the data are reasonably straightforward. Obviously, there are more complex layers to this story that can be developed later, but as a basic observation, we felt that this is relevant and interesting and so we prefer to keep these data in the paper. Some of the experiments have been repeated and new experiments added to make this story more compelling – we hope you will agree. We have omitted the lipid droplet and lipidomics data, as we agree that this appears a diversion.

Detailed Comments

Figure 1 and 1S

1. A significant concern for the entire study is that uridine is not a constituent of the cell culture media. It is well established that a defective respiratory chain impairs cell proliferation. Thus, uridine supplementation in the media is a widely used additive in the culture of mtDNA mutant/patient cell lines. As shown in Figure S1 F1, the growth rate of the patient cell line is lower than the controls. Please clarify the reasons for culturing in this unconventional manner. Also, please see the following landmark papers in the field to support this claim:

doi: <https://doi.org/10.1016/j.celrep.2020.107538>

doi: 10.1073/pnas.1906896116

doi:10.1126/science.2814477 - do these phenotypes exist in the presence of uridine? What is the Akt status like in these cells ± uridine?

We would argue that addition of uridine is not physiological – its addition is a device to improve the growth of Rho0 cells in culture, but we preferred to try to make the situation as physiological as possible, as use of uridine may obscure biochemical adaptations. We note a large number of seminal studies that have not employed additional uridine:

<https://www.pnas.org/content/102/20/7127>

<https://www.pnas.org/content/101/42/15070>

<https://www.sciencedirect.com/science/article/pii/S1097276518300935#sec4>

Nat Chem Biol. 2020 Nov 9. doi: 10.1038/s41589-020-00676-4. Online ahead of print.PMID: 33168978 (this paper only used uridine for Rho0 cells)

In fact, in response to the reviewer's comment, we tried growing cells with uridine supplementation, but found it made no significant difference to growth rates and had no impact on Akt phosphorylation.

2. In Figure 1S, the authors show mtDNA copy numbers for A549 cybrid cell lines. In figure S7A, they follow the time-dependent mtDNA copy number change in different conditions. However, data for mtDNA copy number of the patient cell lines compared to control cell lines is missing.

We have added these data (please see Fig. S1F).

3. Figure 1S shows the membrane potential in A549 cells, but it does not show the morphology data as shown for the patient cell lines in Figure 1F.

We decided to omit the morphological data as this is not an aspect we have explored in depth and it adds little to the narrative.

4. Since the metabolic phenotype presented in Figure S1 does not match or comparable to what they have shown for the patient cells in Figure 1, the use of A549 cells to study the mechanism of 3243A>G cells is unconvincing.

We are confused by this comment: The A549 cybrid cells behaved in every respect in the same way as the fibroblasts (please see Fig. S1G-L). We have reviewed the data and have tried to clarify.

5. The way authors describe patient cells growth rate and mutant cells growth rate (lines 110-112) is confusing for readers. Please state clearly the cell line name as 3243A>G cells or A549 cell line. It should be effortless for the reader to know what they are studying.

We agree and have modified (line 110-114).

6. The NADH: NAD result is very curious indeed. As referenced above, how can there be statistical significance with such experimental variability? This does not compute with such error bars. Please review these data.

These data were statistically significant, have been extended and reanalysed (please see the source data for Fig. 1 and S1).

Figure 2 and 2S

1. The data presentation is meandering. Starting with supplemental figures, returning to the central figure and back again - it becomes irritating and easy to lose interest. The authors are encouraged to think deeply about figure presentation and depict the results in the most straightforward fashion possible.

Much of the analysis is dependent on metaboanalyst, which, although convenient, can vary wildly depending on the selected input parameters. The authors should model the data directly to check if these outputs reflect the best model possible. The metabolomics data from the control samples looks highly variable; why is this? Please address directly.

The data are not highly variable – this may have been a function of the way the data were presented. In fact, they were remarkably internally consistent and reproducible. We have changed the presentation to make this point clear. We have also tried to reorder the figures and text so that it is easier to navigate.

For parameters used for metaboanalyst, we have stated the conditions we used in Methods (line 1137-1139).

2. The authors show two types of patient cells in Figure 1. Which line is used for metabolomics and flux?

We used cells from both patient 1 and 2 for the metabolomics. We have made this clear in the text and figures (please see Fig. 2A and S2A; line 147-148).

3. It is not surprising that lipid droplets form upon respiratory stress. All mitochondrial stressors promote the formation of LDs to some extent. The images are very unusual and look overexposed. Please use a conventional BODIPY dye or omit. I do not see what this data adds to the paper.

We have omitted these data from the paper.

Figure 3

1. This is a very disjointed and confusing figure. Nowhere in the RNA-seq data is autophagy implicated. Everything appears to be very convoluted. Even ULK1 - a master kinase involved in autophagy seems unaltered between the controls and patient fibroblasts in the supplemental data.

We have reorganised the presentation of these data. Actually, RNA-seq analysis showed that a number of genes involved in autophagy, including ULK1, are significantly upregulated in patient fibroblasts (Fig. S4B).

2. Same question as for Figure 2. Which patient cells were used for RNAseq?

Both patient lines were used, as shown in the figures.

3. "These results suggest that the constitutive induction of PI3K-Akt-mTORC1 signalling remodels cell metabolism" is a central thesis of the manuscript. All the raw blotting data should be presented to demonstrate this.

We agree with the reviewer, and have attached the raw blotting data as requested

4. In general, the quality of biochemistry could be much better, and as it stands, it is mechanistically too simplistic. The authors should include full-length blots in supplemental data. The Akt data are very exciting (the most novel part of the paper) but need to be clarified, is this an activating or inhibitory PTM? "Phosphorylation" can mean anything. Is it Thr309 or Ser473? If it is activating, the authors should include monitoring the Akt-dependent phosphorylation of a well-known substrate (e.g. T246 on PRAS40, T346 on NDRG1 or Ser939/Thr1462 of TSC2) to solidify this. Is there a downregulation of a particular phosphatase in the mtDNA patient cell line that could account for this chronic Akt activation? e.g. PP2A (DOI: 10.1126/science.1106148) or PHLPP (DOI: 10.1016/j.molcel.2005.03.008)

We have now included full length blots as requested, and made labelling clearer (please see Fig. S3K and line 212-218)

5. There is no clear rationale for investigating autophagy here, although it is not surprising that it is altered in mitochondrial disease. But this adds nothing to the story. The LC3 immunoblots are not convincing. The levels of LC3 are known to fluctuate between individuals, and thus, inhibitors must be used. Bafilomycin A1 should be used, and all samples should be resolved on the same 13% gel ± starvation.

We have tried to clarify the rationale and the associated data. We have repeated all the LC3B immunoblots with Bafilomycin A1, which yields data equivalent to that obtained using CQ (please see Fig. 3E and S4C-E).

6. As an extension, the quantitation of the blots is problematic. Avoid quantitation by ImageJ/FIJI for western blotting - please use the values provided by the dedicated LiCOR CLx software using total protein normalisation. The ISR data is not convincing and confuses the manuscript once again. ATF4 and particularly CHOP antibodies, whilst widely bought and sold - are notoriously non-specific. Hence the reason for the use of CHOP: XFP reporters by pioneering UPR labs. What does the RNA-seq data say about the ISR - this would be more convincing.

The term "metabolic stress" is vague. Which metabolic stress? It is a respiratory deficiency. The use of the mCherry-GFP-LC3 reporter is more convincing than the biochemical data. However, 20 cells are not sufficient to draw any conclusion. FACS based quantitation of the mCherry-only/total GFP ratio enables quantitation of >10-50K cells. This should be employed ±BafA1 (much more preferable to CQ which is not used much in the field anymore), or high content microscopy analyses (>50-150 cells, dozens of fields) of the reporter should be added to strengthen the data. However, this data is extremely confusing in the manuscript and should really be omitted.

We have reanalysed and represented these data for all the western blotting.

Although RNA-seq suggests inhibited EIF2 pathway and activated ATF4 in patient fibroblasts (Fig. 3A and Table S3), we have omitted the blotting data about the ISR. We think this dataset did not add much to the whole story, since the roles of the ISR in the mutant cells is still not clear in the later part of this manuscript.

The m.3243A>G mutation itself indeed causes a respiratory deficiency. However, based on the results from Figs. 1 and 2, the metabolic remodelling of the mutant cells is far beyond only a respiratory deficiency. That is why we choose to use the term “metabolic stress” rather than “respiratory deficiency”.

We have done some FACS experiments but would argue that the imaging is preferable as we can see exactly what we’re measuring. We have increased the numbers of cells used (> 100 cells) and these data are highly statistically significantly different (please see Fig. 3F and S4F).

7. The muscle biopsy data is excellent! The authors should consider including staining for an Akt substrate (as above) to verify activation at the substrate level in vivo.

We have repeated these and with phospho-S6. However, there is no suitable antibody to PRAS40 and TSC2 (Akt substrates) for immunofluorescence staining.

Figure 4

1. I understand the author's interpretation of the data shown in B, pharmacological inhibition of PI3K-Akt-mTORC1 signalling is in general, affecting the mutant load in A549 cells and patient cells. However, this is a major oversimplification in terms of what we know about the mechanism of Akt signal transduction. The author's interpretation is predicated on the use of LY294002 - widely used but well known non-selective inhibitor of PI3K. LY294002 inhibitors TORC1, PLK1, CK1, PIM1/2, HIPK2 and GSK3 in addition to robustly binding non-kinase ATP-binding proteins (DOI: 10.1042/BJ20061489; DOI: 10.1016/j.chembiol.2004.11.009; DOI: 10.1042/BJ20070797). Accordingly, the authors need to revise these experiments and use much more validated inhibitors for Class I PI3Ks.

Accordingly, the authors should perform revised experiments using allosteric Akt inhibitors known to selectively inhibit phosphorylation of the T-loop/other motifs in Akt, e.g. MK-2206 or AZD5363. For targeted mTORC inhibition, AZD8055 should be used. In light of these comments, all of this data should

be re-examined, and consistent with this, changes in C are not convincing. In the repeat experiments, the authors should include uncropped, full-length blots in supplemental data. The authors should use minimal image processing for western blots in the revised manuscript.

We had already used MK-2206 in the paper, but we have now added experiments with a more specific inhibitor for PI3K, GDC0941. All data proved to be fully consistent and reproducible. For mTOR complexes, rapamycin is known to specifically inhibit mTORC1 rather than mTORC2, so it seemed to us better than AZD8055 which targets both mTOR complexes.

Figure 5

1. The data in Figure 5 are complicated by the absence of uridine and presence of non-selective inhibitors. These data should be repeated to verify the mechanism. The death effects could easily skew the interpretation of load.

We don't understand what is meant by the 'death effects skewing interpretation' – there are minimal death effects and the large volume of work with single cell PCR validate the interpretation of mutant load (please see line 329-338 for the rationale for conducting these experiments).

2. Figure S5 i and ii treatment weeks do not match.

This has now been corrected.

Figure 6

1. Pharmacological inhibition clearly affects the m.3243A>G mutant load, but these effects cannot be attributed to the mechanism postulated due to the specificity of the inhibitors used.

This seems rather harsh – we have seen the same effects with LY, MK and rapamycin everything has been exceptionally reproducible and internally consistent. Inhibition by chloroquine may not be a fashionable way to inhibit mitophagy but its mode of action is not controversial.

2. The mitophagy data are not convincing and confusing as presented. It is hard to know what to focus on with 30 small panels of microscopy images. It is unclear how mitophagy is distinct from alterations in general macroautophagy. It is likely the effects the authors are observing are a function of altered macroautophagy - furthermore, mitophagy can occur independently of LC3.

The Keima quantitation is very confusing, and the quantitation does not reflect the conclusions. Several thousand cells need to be analysed by FACS for mt-Keima due to the variance caused by spectral overlap unless STED imaging/spectral unmixing is employed.

We have increased the sample size and changed the way that these data are presented and hope that this is both clearer and more convincing (please see Fig. 6C and S7G). The dye really does not need spectral unmixing and this has been done using the Airyscan optics and give very clear separation of signals at the two excitation wavelengths.

Figure 7 (mislabelled as S7?)

1. LC3B and CytC colocalisation does not look convincing from the given representative image. These are brief snapshots of a dynamic process and can not be used to infer anything. It does not reflect mitophagy, as this can be independent of LC3.

E compares data from patients from Fig 6C. But the data is represented and quantified in a totally different manner and impossible to state anything. Indeed, the controls look different from these images. Please modify the data in exactly the same style and format in the same panel, or omit these

It is Fig. S7, which corresponds to Fig. 6. For the data of LC3B and Cyt C colocalisation, we have replaced the data with Flow cytometry analysis for cybrid cells transfected with COX8-mCherry-EGFP (please see Fig. S7F), which should directly reflect mitochondrial degradation and thus be more convincing. For the former S7E, we have revisited these data, presenting them in a different way (please see Fig. 3G) and hope that this will prove more acceptable.

Reviewer #2 (Remarks to the Author):

The paper by Chung et al., reports on studies in fibroblasts and cybrid cells carrying the m.3243A>G mutation. The authors have extensively characterised changes in bioenergetic, metabolomic, lipidomic and RNAseq profiles in heteroplasmic patient-derived cells and revealed that the mutation promotes upregulation of PI3K-Akt-mTORC1. Pharmacological inhibition of PI3K, Akt, or mTORC1 activated mitophagy, reduced mtDNA mutant load and rescued cellular bioenergetics cell-autonomously, suggesting that activation of the PI3K-Akt-mTORC1 pathway represents a potential therapeutic target. I have a few questions:

1. On Figure 1 the RFLP shows much weaker band for the mutant than for the wild type. How was the 79% heteroplasmy calculated?

The number, 79%, is obtained using ARMS-qPCR, which is a quantitative way to measure heteroplasmy. More specifically, mutant level (%) was calculated using $1/[1 + (1/2) \Delta CT] \times 100\%$, where ΔCT (cycle threshold) = $CT_{\text{wild-type}} - CT_{\text{mutant}}$. In contrast, PCR-RFLP is relatively a qualitative way to detect mtDNA mutations. Because of an intrinsic problem of PCR-RFLP, part of the PCR products are hybrids which have a strand of wild-type (WT) and a strand of mutant (a single nucleotide mutation is not strong enough to avoid their annealing). Moreover, restriction enzymes cannot recognise and cleavage these hybrids of WT/mutant DNA, which means that in the “WT” band, part of the PCR products are hybrids. Thus, the PCR-RFLP largely underestimates the mtDNA mutant loads, which is the reason why there are weaker bands of the mutant than which of the WT.

To avoid misunderstanding, we have added a few more sentences in Results and Methods to explain that (line 81-83 and line 919-942).

2. What was the rationale behind the reduced Glutamine in the medium? In Figure 1 there is no change on growth rate of low Glutamine, but on Figure 1S on 0.1 mM Glutamine there was a significant growth defect. Please, explain.

1) Glucose and glutamine are two major carbon sources for cell metabolism. Since the mutant cells have dysfunctional mitochondria, we wondered whether the cells rely primarily on glycolysis or on glutaminolysis for bioenergetic support. We have revised the sentence (line 107-108) to explain the rationale.

2) Glutamine addiction is a general feature of cancer cells, which means that they need high concentrations of extracellular glutamine for growth. This may explain that the cancer cells, A549, and A549 derived cybrid cells are more sensitive to glutamine deprivation. Also, the time scale for quantifying cell growth is only 48h and might not be long enough for fibroblasts to show growth defect.

3. There was an increase in some amino acids, potentially due to increased uptake. Was there any change detected in the amino acid transporters of these amino acids?

RNA-seq analysis showed that amino acid transporters of Glu, Asp, Cys, Arg, Lys and His are upregulated, while Gln transporter is downregulated in the A3243G fibroblasts (please see Fig. S3H and the following Table).

Genes	padj (Pats vs Ctrls)	Gene function
SLC1A1	0.00	L-Glu, D/L-Asp, L- Cys; Learning, memory, motor neuron behavior, GABA and glutathione synthesis, protection from oxidative stress
SLC6A7	0.21	Glutamatergic neurons; L-Pro; Presynaptic regulatory role in excitatory synaptic transmission
SLC7A11 (xCT)	0.10	L-Glu, L-Cys; L-cystine uptake, glutathione (GSH) synthesis
SLC15A4	0.00	L-His; L-His and oligopeptide transport from inside the lysosome to the cytosol, production of interferon type I, endolysosomal pH regulation
SLC38A4	0.03	L-Gln, L-Arg; Insulin secretion, glucose recycling, nitric oxide synthesis
PQLC2	0.00	L-Arg, L-Lys, L-His, L-canavanine, L- Orn, MxD (cysteamine and cysteine mixed disulfide); Lysosomes
SLC36A1	0.27	GABA; Heme biosynthesis, transport of aminolevulinic acid Heme biosynthesis, transport of aminolevulinic acid
CTNS	0.03	L-cystine, L- selenocysteine, L- cystathionine; L-cystine efflux from lysosomes
SLC25A38	0.00	L-Gly; Heme biosynthesis
SLC38A7	0.12	L-Gln, L-Ala; maintenance of L-Glu neurotransmitter pool in the brain through the glutamine-glutamate cycle
SLC3A2	0.57	
SLC7A6	0.27	CAAs and LNAAs; L-Arg transport in astrocytes, NO synthesis
SLC6A17	0.69	
SLC38A9	0.09	L-Arg, L-Leu; mTOR activation
SLC7A8	0.82	
SLC25A22	0.58	L-Glu; Transport of L-Glu across the IMM into the mitochondrial matrix
SLC38A10	0.53	
SLC17A7	0.92	L-Glu; Presynaptic vesicular L-Glu packaging in cortical neurons, glutamatergic chemical transmission, insulin secretion
SLC38A1	0.94	L-Gln; Insulin-regulated glucose metabolism, mTOR activation
SLC38A2	0.88	L-Gln; mTOR activation, embryonic development, glutaminolysis
SLC25A12	0.87	L-Glu, D/L-Asp; ATP synthesis, supply of precursor for malate-aspartate shuttle, mitochondrial respiration, calcium signaling and antioxidant defenses
SLC1A5	0.98	L-Asp, L-Cys, L-Gln; L-Gln transport, mTOR activation, and supply of carbon source for TCA cycle through L- Gln transport
SLC25A13	0.92	L-Asp, L-Glu; Transports cytosolic reducing equivalents produced during glycolysis into the mitochondria as part of the malate-aspartate NADH shuttle, Redox homeostasis and the calcium- mediated control of mitochondrial respiration
SLC7A1	0.99	CAAs (L-Arg)
SLC36A4	0.97	L-Pro, L-Trp
SLC1A4	0.99	L-Ala, L-Ser; L-Ser transport from astrocytes to neurons, neuronal development and function
SLC6A15	0.91	BCAAs; Food intake, body weight
SLC7A7	0.28	Cationic and NAAs; Intestinal absorption and renal reabsorption of the CAAs L-Orn, L-Arg and L-Lys

SLC1A3	0.06	L-Glu, D/L-Asp; L-Glu clearance from synaptic cleft
SLC43A1	0.08	BCAAs; mTOR activation, Regulation of cellular thyroid hormone levels
SLC15A3	0.40	L-His; Reabsorption of dietary peptides in the intestine and kidney and maintaining homeostasis of neuropeptides in the brain
SLC25A15	0.01	L-Orn, L-Cit; L-Orn transport, Urea cycle and mitochondrial protein synthesis
SLC7A5	0.54	LNAAs; L-Leu transport, mTOR activation, pro-inflammatory cytokine production, pancreatic b-cell signaling and function
SLC1A2	0.27	L-Glu, D/L-Asp; L-Glu clearance from synaptic cleft, brain development
SLC38A5	0.00	L-Gln; Glutaminolysis, glutamine- glutamate cycle
SLC38A3	0.02	L-Gln; L-Gln release from astrocytes and pancreatic b-cells, glucose homeostasis, renal ammoniogenesis
SLC7A14	0.00	CAAs; Retinal development and visual function

4. How can the authors prove that the PI3K-Akt-mTORC1 signalling is the cause of the metabolic remodelling?

We apologise if the text was misleading. We accept that we did not prove that the PI3K-Akt-mTORC1 signalling is the cause of the metabolic remodelling, although we proposed that they are highly correlated. It is still hard and will take a long time to dissect the causality between cell signalling and metabolic remodelling in the scenario of mtDNA mutations, which is beyond the scope of this paper. To avoid misunderstanding, we also emphasize that “the causality between the observed alterations in metabolism and cell signalling of the m.3243A>G mutant cells is not yet conclusive” in the last paragraph of Discussion (line 508-509).

5. While some papers suggest that the activation of the ISR is protective and promote longevity, the data in this paper suggest that the activation of ISR in the context of the m.3243A>G mtDNA mutation is not protective. How can the authors be sure that it is not triggered a protective compensatory mechanism (e.g. metabolic rewiring)?

We agree that this is an interesting question. We are not sure how best to answer this based on the data we have now. Other reviewers also suggest omitting this dataset. We therefore decided to remove the data of immunoblotting for proteins involved in ISR, since we did not further explore the roles of ISR in cells bearing the mutation in this paper.

6. In cell culture it is possible that the mutation load is changing due to the predominant loss of cells with high rate of mutant and defective OXPHOS, and better survival of cells with low or no mutation rate. How can the authors be sure that the treatment caused the reduction of mutation rate? This rate may also be a bit dependent on the starting heteroplasmy rate. It would be good to have some comment on this.

This was a concern for us, and was the reason that we conducted long-term cell growth/death measurements and single-cell qPCR to exclude the possibility (Fig. 5 and S6; line 327-374).

7. How can the authors explain the difference of pathways involved in cells carrying the m.8993T>G mutation and the m.3243A>G mutations? Is the defect detected in m.3243A>G more similar to other defects of mitochondrial protein synthesis? Or this would be specific for m.3243A>G only?

This is a fundamental and important question which warrants further exploration. While we cannot yet explain the difference, we think it is an important and interesting finding. We would have to carry out extensive comparative cell biology across a number of mutant lines to establish whether this is specific to the m.3243A>G mutation.

8. Is the improvement only linked to reduced heteroplasmy rate? Can it be influenced by a better metabolic remodelling of metabolism?

As aforementioned in comment 4, We are not yet clear about the causality between cell signalling and metabolic remodelling and, yes, it is possible that the improvement could result in part from a better metabolic adaptation. However, due to the difficulty and time for studying this question, it is beyond the scope of this paper in which we have tried to focus on the rescue of mutant load.

9. Immunofluorescence staining revealed a significant decrease in MT-COI expression in patient 1 fibroblasts compared to controls. Was this result only in the cells with higher heteroplasmy rate? It has been reported that the m.3243A>G mutation is affecting more complex I than complex IV in muscle biopsies, so it would be understandable if MT-COI would be only reduced in the more severely affected cells.

The reports vary. Some suggest that a complex IV is most affected, probably because it's the most abundant subunit expressed in the ETC. We observed MT-COI difference only in fibroblasts from patient 1 but not patient 2.

10. The mtDNA copy number in the mutant cells was very highly increased (see Fig. S1 A), suggesting a compensatory mitochondrial biogenesis with mtDNA replication. Is it possible that the wild type mtDNA molecules replicated faster?

The increase of mtDNA copy number was only seen in the cybrid cells but not patient-derived fibroblasts. It is possible that mutant mtDNA replicate faster than WT and some reports also support this hypothesis. Our data in this manuscript showed that the total mtDNA copy number remained stable and mitophagy was upregulated across the duration of the experiment, while blockade of mitophagy using chloroquine prevented the decrease of mutant loads. This suggests that mitophagy play a key role in removing mutant mtDNA. On the contrary, if WT mtDNA replicated faster than mutant ones after drug treatments, we should be still able to see a decrease in mutant loads when the mutant cells were treated with chloroquine because chloroquine only blocks mitophagy and will not affect mtDNA replication.

11. Published RNA-seq data set of muscle biopsies from patients with MELAS detected increase in FGF21 (~48 fold) and GDF15 (~9 fold), these are also present in serum of these patients and therefore suggested to be biomarkers of mitochondrial translation defects. In contrast, in the present study no changes in these two genes were detected in the patient fibroblasts. How can the authors explain this? Are fibroblasts a good model of the metabolic situation in muscle?

We accept that fibroblasts are not a perfect model, and we are working on developing a muscle model, but this in itself is a challenge. However, we believe that there should be a general change in response to the mtDNA mutation no matter what cell types they are. The secretion of FGF21 and GDF15 could be a cell-type specific response and a major phenotype seen in the tissues such as muscles sensitive to amino acid starvation. This is also observed in the RNA-seq data using patients' muscle biopsies (Table S4). Moreover, we also observed an increase in FGF16 (~33 fold) in fibroblasts (line 483). Thus, data from fibroblasts and from patients' muscle biopsies seem to be consistent.

12. Some of the western blots in Figure 3 are not really showing the significant defect which is highlighted on the quantification graphs. The blots were repeated for the quantifications. Are there any of the blots better to demonstrate the changes?

We have revisited and represented these data (please see Fig. 3B and its WB raw images).

Reviewer #3 (Remarks to the Author):

1., 'the authors use a pharmacological inhibition of PI3K, Akt or mTORC1 and show a significant improvement of practically all studied phenotypes. This is clearly a highlight of the study; however, also the most problematic point, as the authors do not compare these rescues in patient cells to control cells and/or control cells treated with the same drugs. Thus, the reader gets an impression that everything is rescued, although there still might be a significant biochemical defect present. What is the effect of the PI3K-Akt-mTORC1 axis inhibition by rapamycin or LY294002 on control and A549 cells? This major issue needs to be rectified. Regardless of the fact that the rescue might be only partial, the presented results are of great importance to the mitochondrial research field'.

We have now included comparisons of measurements from control or WT cells for all manipulations following drug treatments to show the absolute levels of rescue – in some measures these are complete, in others partial, but these are now transparent in the data as presented (see Fig. 4C-D, 4H-I, S5D-E, and S5H-I). We have also extended the inhibitors as requested by another reviewer to include a more selective PI3K inhibitor and an AKT inhibitor, expanding these data all of which remain fully internally consistent and reproducible

2. - *Although the mutation load is different for patient 1 and 2 (86% and 30%, respectively), there were very little differences in the biochemical phenotype between them? What is known about the threshold for this mutation in MELAS patients? Would it be expected that 30% mutation load is enough to cause such a defect?*

We were also surprised by the bioenergetic defect in patient 2 given the relatively low mutant load. This applies more to some measures (oxygen consumption) than to others (lactate generation and membrane potential). That said, the subject who donated the cells is ill, with a cognitive and psychiatric problem and muscle weakness. All we can do here is to report the data as we found it.

3. *Also, the mutant load in patient fibroblasts and cybrids reported in Fig. S6B is different from the mutant load reported previously throughout the manuscript. I gather the techniques used are different, but it is worrisome that the precision of at least one of these techniques is not good. Please, comment.*

The precision of the measurements is fine – the mutant load does tend to drift with time in culture and these differences simply reflect that. We have tried to clarify in the text.

4. *What is the reasoning for the increase of the mutation load of patient 2 over time?*

We don't know why the cell lines behave differently. We can but speculate that activation of the PI3K-Akt-mTOR pathway is permissive for expansion of the mutant load in cells from 'patient 2', as inhibition of the pathway limits accumulation of the mutant.

Similar phenomena have been observed in other cell models with mtDNA mutations.

Ref: <https://www.pnas.org/content/110/38/E3622>

5. *Why was the mtDNA increased 4-5 times in the mutant cybrids? Was the mitochondrial content (measured by enzyme activities/protein levels) changed? Why was the OCR normalized to the mtDNA level? Were all the other results normalized to mtDNA level? What was the OXPHOS supercomplex assembly in these cells? Please, explain the rationale and comment to this in the manuscript.*

The cybrid cell model is quite an artificial model, and we were not terribly surprised that the mtDNA content should be different from the native cells. The mitochondrial content was similarly increased as seen for example in imaging and protein content on WB. Given this huge difference in mtDNA copy number, it seemed necessary to normalise the respiratory rates which reflect the total mitochondrial mass. mtDNA copy number seemed to us the most reliable measure to use to be quantitative. There is no need to normalise most other data to mtDNA copy number – for example measurements of potential made by confocal imaging are independent of mitochondrial mass. As supercomplex assembly is measured from mitochondrial preps and is a function of protein concentration, it is also independent of mtDNA copy number. These were quantified by normalisation to a nuclear encoded mitochondrial protein (ATP5A) which also solves that problem.

6. *Fig. 1E & 4I – the levels of Complex II are also affected in the patient cells, however subunits of this complex are encoded by nuclear genome, and in general Complex II is not affected in patients with mitochondrial mutations, rather often used as a loading control for BN-PAGE? Please, explain.*

We were also surprised by the decrease complex II, but these were the data we obtained, and they suggest that the use of complex II to normalise data should be treated with greater caution.

7. *There are striking differences in the metabolomics, lipidomics and transcriptomics of the two controls used in the study. This is probably a normal situation, as the cells are not isogenic, but it makes it difficult to compare them to the results in the patient cells. The authors should comment on this in the text.*

We have removed the lipidomics from the paper. However, we disagree – the differences in most variables between the two patient lines and the cybrid cells were surprisingly consistent and clearly distinguished from controls.

8. *Fig. 2C, 3G – the representative WB of patient and/or cybrid cells do not really correspond to the quantification, and in other cases immunoblots are not shown. The visualization of the WB is problematic throughout the manuscript, could the authors improve the signal-to-noise ratio, or show them all in colour?*

We have revisited and represented these data in grey scale (please see Fig. S2D and 3B and their WB raw images)

9. Line 202 - Specifically, the increased expression of PC was consistent with the immunoblotting data (Fig. 2C) and correlated with altered TCA cycle activity (Fig. S3G) – this is valid only for patient 2, please comment/correct.

We have corrected that (line 171-172).

10. *Fig. 3H – muscle biopsy does not look very good, could an H&E stain be provided to determine that*

no freezing issues are creating this pattern? Also, have the antibodies used here been tested to indicate that the staining is specific?

We have included an H&E and also a histochemical stain of COX1 in Fig. S4A and we have replaced the image used (now Fig. 3C). The biopsies and sections are fine. Antibodies are all validated in knockouts etc.

11. It is interesting to see that the inhibition of PI3K-Akt-mTORC1 axis in m.8993T>G mutant cells had no effect on the mutation load; and highlights the importance, as mentioned above, to study individual mutants and their effect on cellular metabolism. Did you assess the phosphorylation status and total levels of PI3K-Akt-mTORC1 axis proteins and the downstream reporters of ISR (ATF4, ATF5, CHOP) in m.8993T>G mutant cells and compare them to your cybrids?

Agreed – we think this is an important finding. Yes, we measured AKT and S6 phosphorylation and LC3BII to I ratio in these m.8993T>G mutant cells with mutant loads of 0%, 50%, and 80% and found no difference (please see Fig.S7H). We didn't follow up with ATF4/5 or CHOP, given that the upstream signals were not changed.

12. - Fig. 5B – the cell growth was significantly increased in patient cells treated with rapamycin, however the text states that this is not the case. Please, clarify.

We have added a few more sentence to describe and explain the result (line 341-345)

13. - Fig. S6D – the figure legend talks about COXIV quantification, however the main text and figure mentions mt-COI? Which protein was quantified? Also, in (ii) and (iii) the patient column is identical, and the two graphs should be merged together and statistical analysis should be done comparing treated patient cells and controls (as mentioned above). (iv) and (v), patient curve is identical, again the graphs should be merged into one.

We thank the reviewer for spotting this error. MT-COI was measured in the immunofluorescence (now in Fig. S6E). We have also represented the data according to the reviewer's suggestion.

14. - It has been shown that LC3 might aggregate in an autophagy-independent manner, thus colocalization with mitochondria might not signify an increase in mitophagy, especially when control cells are not shown. Thus, the use of mt-Keima became the golden standard in determination of mitophagy. Here, however, 543nm-mt-Keima is very strong and practically labels all mitochondria in both patient and control cells? Addition of chloroquine should diminish the 543nm-mt-Keima signal, as the fusion of lysosome and autophagosome is inhibited by it, but this is not the case. It is difficult to assess the level of mitophagy based on the picture provided. Please provide better images.

We have revisited the images and provided better images in Figs. 3G (Ctrls vs Pats), 6C (Pat 1 vs LY/RP treated Pat 1) and S7G (Pat 1±LY/RP+CQ).

15. - Although a possible activation of mtUPR is suggested, no data are shown. Was there any activation of this pathway in the transcriptomics analysis? Did the authors look at the protein levels of mtUPR markers?

Thank the reviewer for this question. As shown in Fig. 3E, RNA-seq analysis suggests the UPR pathway is enriched. However, we did not measure the protein level of UPR^{mt} markers. Moreover, another reviewer has suggested omitting the dataset about ISR, since we did not explore the roles of ISR/UPR^{mt} in cells bearing the mutation in this paper. We therefore decided to remove the data

of immunoblotting for proteins involved in ISR and only discuss the potential roles of ISR/UPR^{mt} in Discussion (line 464-484).

Minor comments/questions:

1. - Scale bars are sometimes missing, especially in zoomed images.

Scales bars have been added to all images, please see Figs. 3F, 6B, S4F, and S7C.

2. - Fig. S1J - the differences between WT and Mutant are statistically different, but the difference is minor. Could this be caused by the fact that 180 cells were counted, where in other experiments only 50-80 cells are counted?

Yes, the N numbers affect the statistic power. However, although the difference is minor, we think it is still meaningful, because the ratio of NADH:NAD⁺ is tightly regulated and is a balance of several metabolic pathways, including glycolysis, OXPHOS, malate-aspartate shuttle, etc. Take a paper titled “NADH Shuttling Couples Cytosolic Reductive Carboxylation of Glutamine with Glycolysis in Cells with Mitochondrial Dysfunction” as an example. The measurement of NAD⁺:NADH in Fig. 3A of the paper showed less than 10% difference compared to control. In contrast, the difference of NADH:NAD⁺ ratio between WT and the m.3243A>G cells in our manuscript was about ~20%, which is a relatively large difference.

3. - Fig. 2D a lot of the fluorescence is outside of the cells, how was the cell area defined, the figure should clearly show the cell outline, otherwise it is meaningless.

We have omitted the lipid droplet data, as we agree with another reviewer that this appears a diversion.

4. - glycerol-3-phosphate is sometimes annotated at G3P, αG3P or alphaG3P - please unify

We are sorry for the confusion and have corrected that (please Fig. 2B and line 156).

5. - Line 339 - MK showed similar results in terms of cell growth and death (Fig. S5A). – Should be Fig. S6A

We thank the reviewer for spotting this error and have corrected that (line 348).

6. - Line 509 - (A) PCR-RFLP and ARM-qPCR - should be ARMS-qPCR

We thank the reviewer for spotting this error and have corrected that (line 525).

7. - Fig. 5 – please explain pre-treatment/treatment in this experiment

In this experiment, cells were pre-treated with either vehicle, LY, or RP for time periods as specified in Fig. 5B (pretreatment) and then seeded into 96-well plates for measuring cell growth rate and cell death with either vehicle, LY, or RP (treatment). We have modified Fig. 5A and added an explanation for these two terms in the text (line 335-338) and the figure legend (line 776-784).

REVIEWER COMMENTS

Reviewer #1 (Remarks to the Author):

The authors have revised the manuscript and satisfied several concerns. The omission of LD data was a step in the right direction, but the paper is still difficult to read. A central issue is still the assumption that defective degradation is the mechanism. The weakest aspect of this manuscript is the data referring to "mitophagy/autophagy". Without the focus on autophagy or mitophagy, this is a genuinely intriguing story with significant translational relevance. With this data, however, the paper is confusing, cluttered and an otherwise pertinent and nice observation is lost(chronic AKT activation, which is novel). This reviewer would support the publication of the manuscript, only without the autophagy and mitophagy data as much more work is needed to resolve several issues.

To clarify, the authors have not explored the regulation of either macroautophagy or mitophagy in sufficient depth, and the phenotypes presented are incredibly subtle - yet major claims are made. There are also questionable methodological "red flags" from reading the methods and legends, in concert with some unique interpretations. Thus, the data do not reflect the claims.

In the interests of providing a constructive rationale for this viewpoint - the author's claims are based on the invocation of the following data:

1. RNA Seq data in Figure 3Ei – which the authors claim mTOR signalling is justification for autophagy. This interpretation is not a conventional rationale used in the autophagy field to focus on autophagy.
2. Induction of "autophagy genes" in Fig S4Bii – this heatmap depicts variability and not a consistent autophagy signature in the first column (hardly any difference when all the data are assessed - all pats vs all controls).
3. LC3B blotting. The critical data on LC3BII are not convincing (as would have been predicted from RNA-seq). In accordance with the well-established conventions in the field, LC3-I levels are known to vary wildly between cells, individuals and tissues and thus cannot be used as a rigorous readout of any autophagic activity. The only result in the context of LC3 that matters is its conjugation to PE, i.e. conversion of LC3-I to LC3-II. This is not affected in their cells even with lysosomotropic inhibitors, demonstrating no effect of autophagy here. The authors are strongly encouraged to consult a classic in the field - PMID: 17611390. Contrary to the author's impressions, this does not indicate any defect at a late stage. For this, verification by additional pulse-chase assays through the endolysosomal system would need to be performed. The data in Figure 3Eii and iii indicate no difference in autophagy. There

are no immunostains to autophagy markers at defined stages to support the author's assertions. There is no data on ATG5-12 conjugation, p62 accumulation or otherwise. There blots and data presented do not reveal any discernable defect.

4. The author's data on macroautophagy and mitophagy reporters and their interpretation are inconsistent with standards and conventions of the field.

"Consistent with the above result, the ratio of green/red puncta was low in control compared to patient cells, representing the formation of mature autolysosomes and a high autophagy flux. In contrast, the ratio of green/red puncta and both the number and size of double-positive puncta are increased in mutant cells, indicating impaired autophagosome maturation into autolysosomes (Fig. 3F)"

- This is not a conventional or typical interpretation – reporters do not give this level of insight or resolution. The major red flag here is the appearance of the reporter in the control images, which exhibits a very unusual signal and pattern. This reviewer predicts there has been an issue in the transfection of cells with this construct, as the characteristic diffuse green signal is not visible. Either the cells have been generated inappropriately, or the image acquisition settings are unorthodox. There should be a diffuse green signal in the control cells and hardly any red-only puncta. Indeed, these images of the LC3 reporter do not reflect the blotting data and look problematic. Please see PMID 20144757.

5. The quantitation is also problematic. Are these significance values derived from all of the values in all experiments – which would lead to significance, or from the mean values of each biological experiment?

6. Transient transfection is problematic for autophagy reporters, particularly if using cationic lipid methods such as lipofectamine, which are well known in the autophagy field to induce/affect LC3 lipidation and autophagy - PMID: 20383065. Viral mediated delivery and stable cell line generation is the most robust method. Thus, the structures detected and quantified cannot be regarded with high confidence.

7. The mechanisms of CQ and BafA1 are incredibly different. Both inhibit flux, but BafA1 is a lysosomal vATPase inhibitor, CQ is a membrane fusion inhibitor that does not affect vATPase function.

8. Based on the above data from macroautophagy, the rationale for investigating mitophagy is still unclear. Mitophagy is a very different process from bulk macroautophagy – although macroautophagy can, in some instances, result in high levels of mitochondrial elimination - revealed by using both reporters. This is not traditional "selective autophagy" however i.e. mitophagy. Its just increased macroautophagy and a large number of mitochondria are also turned over as a consequence.

9. The mitophagy data itself is unusual. According to the methods section, laser settings differ "between each experimental condition" for imaging of mt-Keima. This is not standard practice and explains the discrepancy. Control fibroblasts have incredibly high levels of baseline mitophagy, which is unusual. I predict that with the author's experimental setup, i.e. transient transfection using lipids and differential laser settings, they would see the same result for any organelle labelled with this reporter.

There also appear to be aberrations within the images (a clear case of z-drift or misalignment in the photo presented in Figure 3Fi). From the extensive use of such, they should not look like this - especially with Airyscan or other high-quality imaging setups.

10. The authors have an outdated understanding of the field and assert "mitophagy is absolutely required" – and nothing else. Macroautophagy and mitophagy are very different processes. An increase in macroautophagy means elevated non-selective bulk turnover. Thus, using a mitophagy reporter – their increase in the signal can be attributed to this. However, it does not mean that the selective autophagy of mitochondria is actively occurring (i.e. the definition of mitophagy). There is no data on the receptor mediating the turnover of mitochondria or how its induction would differ from macroautophagy. As it stands, these terms are collectively bundled together and used interchangeably, and it isn't very pleasant to read. There is a wealth of literature on this that should be consulted.

11. The authors mention that mitophagy has been studied in the context of combined antimycin/oligomycin treatment. Yet no further information is provided. This would have serious implications for mechanistic understanding if mitophagy was in fact affected.

12. How does chronic AKT activation lead to mitochondrial elimination, and how is this different from macroautophagy?

There are also several confusing statements concerning autophagy in the paper: "suppresses macromolecular catabolism by autophagy" – this sentence is confusing. Autophagy is, by nature, a self-catabolic process. The induction of macroautophagy is the induction of non-selective catabolism.

In terms of the rebuttal letter, the Senior author writes that the "Inhibition by chloroquine may not be a fashionable way to inhibit mitophagy but its mode of action is not controversial". This reviewer would like to highlight that this assumption is, in fact, very incorrect. The mechanism of CQ and HCQ was actually a contentious issue in the autophagy field until it was finally resolved that it inhibits fusion without any effect on lysosomal pH (PMID: 29940786 - almost 700 citations since 2018).

This reviewer thinks that the manuscript is exciting but is undermined the problematic focus on autophagy. Without this data, the story would flow and make sense, and this reviewer would support its

publication. In its current form, the autophagy data has been acquired and interpreted strangely. This does not mean these processes are not involved (although there are no differences in the key assays). The discovery of chronic AKT activation in mitochondrial disease is novel and would merit publication. However, far more compelling and rigorous data would be required to justify any credible invocation of macroautophagy or mitophagy here. The current data do not support this as a mechanism.

Reviewer #2 (Remarks to the Author):

I think the revised manuscript has improved. I do not have more questions.

Reviewer #3 (Remarks to the Author):

The authors have addressed majority of my concerns; however, I would still argue that the inhibition PI3K-Akt-mTORC1 axis results in a partial rescue of the cellular bioenergetic function and the authors should report the rescue as “partial” throughout the manuscript, including the abstract. The manuscript is well written and easier to read in this form after leaving some data out, yet some small issues need to be addressed before acceptance:

The authors show a high mitophagic flux in control fibroblasts in Fig. 3G/H? Why do so many control cells undergo mitophagy? Were these cells somehow treated to induce mitophagy? What happens if mitophagy is inhibited by CQ or Bafilomycin?

The treatment of patient and cybrid cells with inhibitors of PI3K-Akt-mTORC1 axis, clearly improved the OXPHOS capacity, cells growth and supercomplex assembly, however as no treatment of control cells was done in parallel, it is impossible to assess if the partial rescue was achieved due to an overall mitochondrial biogenesis? The increase of Complex II suggests that the used drug might have such an effect. Please, comment.

Fig. 5B is very difficult to understand. What shows the pretreatment and what shows the treatment? Is it possible to change the graphs and group 0-pretreatment together and show within that group the effect of the treatment, then 4-weeks pretreatment, etc.? Also, how long was the treatment/measurement? Is it possible that all the high mutant load cells die within the first 24hours, and the expansion of lower mutant cells is propagated further? Would it be possible to pick a clone with

a high load (at least in the case of the cybrids) and perform this experiment on a clone, to really prove this point? SytoGreen stain labels both live and dead eukaryotic cells, how was it used to measure dead cells?

Fig. 6A - The autophagic flux is always increased by CQ treatment (e.g. Fig. 3E), as it prevents conversion to autolysosome. The LC3BII/LC3BI ratio between untreated and LY or RP treated cells should be compared to assess the autophagic flux, and this is not significantly increased based on this figure.

Fig. 6B, C – treatment of patient cells with RP and LY restores autophagic and mitophagic flux within 24 hours, however the decrease in mutant load does not happen until 4 weeks of treatment. If mitophagy starts happening at this point, why does the mutant load persist? Is mitophagy a very inefficient process?

Minor comments:

Fig 3C-D - only one patient biopsy was used, however the figure legend and in D) figure description gives an impression that multiple patient biopsies were used: it should be corrected. What % of fibers was p-Akt/p-S6 positive in the patient biopsy?

Fig. 3E, S4C, 6A, S7B/E/H - two different sets of samples are shown, however only one loading control set. Is the loading control of cells treated or untreated with a drug? Please, provide a loading control for both sample sets. If only LC3BII/LC3BI ratio is calculated, the loading controls are not necessary, and could be removed?

Line 236 – “showed an increased conversion to LC3BI to the lipidated LC3BII form”, should be ...”conversion of LC3BI to the lipidated LC3BII form”

Fig. 5C-D – in the main text the authors talk about a treatment of 6 and 12 weeks, however in the figure legend it is 4 and 8 weeks. Which is correct?

Discussion – “In contrast, in the present study, while we did not find changes in these two genes in the patient fibroblasts, we did find upregulation of FGF16 (~33 fold), suggesting that changes in FGF pathways in response to mitochondrial dysfunction might be tissue specific.” Was FGF16 increased in the cybrids? Please, correct the sentence.

REVIEWER COMMENTS

Reviewer #1 (Remarks to the Author):

The authors have revised the manuscript and satisfied several concerns. The omission of LD data was a step in the right direction, but the paper is still difficult to read. A central issue is still the assumption that defective degradation is the mechanism. The weakest aspect of this manuscript is the data referring to "mitophagy/autophagy". Without the focus on autophagy or mitophagy, this is a genuinely intriguing story with significant translational relevance. With this data, however, the paper is confusing, cluttered and an otherwise pertinent and nice observation is lost (chronic AKT activation, which is novel). This reviewer would support the publication of the manuscript, only without the autophagy and mitophagy data as much more work is needed to resolve several issues.

To clarify, the authors have not explored the regulation of either macroautophagy or mitophagy in sufficient depth, and the phenotypes presented are incredibly subtle - yet major claims are made. There are also questionable methodological "red flags" from reading the methods and legends, in concert with some unique interpretations. Thus, the data do not reflect the claims.

We have decided to accept the reviewer's advice and have removed all reference to data about autophagy/mitophagy. As all the reviewer's comments refer to the autophagy issue, we assume that these comments no longer require a response. It seems we have a lot to learn about mitophagy and macroautophagy.

In the interests of providing a constructive rationale for this viewpoint - the author's claims are based on the invocation of the following data:

1. RNA Seq data in Figure 3Ei – which the authors claim mTOR signalling is justification for autophagy. This interpretation is not a conventional rationale used in the autophagy field to focus on autophagy.

2. Induction of "autophagy genes" in Fig S4Bii – this heatmap depicts variability and not a consistent autophagy signature in the first column (hardly any difference when all the data are assessed - all pats vs all controls).

3. LC3B blotting. The critical data on LC3Bii are not convincing (as would have been predicted from RNA-seq). In accordance with the well-established conventions in the field, LC3-I levels are known to vary wildly between cells, individuals and tissues and thus cannot be used as a rigorous readout of any autophagic activity. The only result in the context of LC3 that matters is its conjugation to PE, i.e. conversion of LC3-I to LC3-II. This is not affected in their cells even with lysosomotropic inhibitors, demonstrating no effect of autophagy here. The authors are strongly encouraged to consult a classic in the field - PMID: 17611390. Contrary to the author's impressions, this does not indicate any defect at a late stage. For this, verification by additional pulse-chase assays through the endolysosomal system would need to be performed. The data in Figure 3Eii and iii indicate no difference in autophagy. There are no immunostains to autophagy markers at defined stages to support the author's assertions. There is no data on ATG5-12 conjugation, p62 accumulation or otherwise. There blots and data presented do not reveal any discernable defect.

4. The author's data on macroautophagy and mitophagy reporters and their interpretation are inconsistent with standards and conventions of the field. "Consistent with the above result, the ratio of green/red puncta was low in control compared to patient cells, representing the formation of mature autolysosomes and a high autophagy flux. In contrast, the ratio of green/red puncta and both the number and size of double-positive puncta are

increased in mutant cells, indicating impaired autophagosome maturation into autolysosomes (Fig. 3F)"

- This is not a conventional or typical interpretation – reporters do not give this level of insight or resolution. The major red flag here is the appearance of the reporter in the control images, which exhibits a very unusual signal and pattern. This reviewer predicts there has been an issue in the transfection of cells with this construct, as the characteristic diffuse green signal is not visible. Either the cells have been generated inappropriately, or the image acquisition settings are unorthodox. There should be a diffuse green signal in the control cells and hardly any red-only puncta. Indeed, these images of the LC3 reporter do not reflect the blotting data and look problematic. Please see PMID 20144757.

5. The quantitation is also problematic. Are these significance values derived from all of the values in all experiments – which would lead to significance, or from the mean values of each biological experiment?

6. Transient transfection is problematic for autophagy reporters, particularly if using cationic lipid methods such as lipofectamine, which are well known in the autophagy field to induce/affect LC3 lipidation and autophagy - PMID: 20383065. Viral mediated delivery and stable cell line generation is the most robust method. Thus, the structures detected and quantified cannot be regarded with high confidence.

7. The mechanisms of CQ and BafA1 are incredibly different. Both inhibit flux, but BafA1 is a lysosomal vATPase inhibitor, CQ is a membrane fusion inhibitor that does not affect vATPase function.

8. Based on the above data from macroautophagy, the rationale for investigating mitophagy is still unclear. Mitophagy is a very different process from bulk macroautophagy – although macroautophagy can, in some instances, result in high levels of mitochondrial elimination - revealed by using both reporters. This is not traditional "selective autophagy" however i.e. mitophagy. Its just increased macroautophagy and a large number of mitochondria are also turned over as a consequence.

9. The mitophagy data itself is unusual. According to the methods section, laser settings differ "between each experimental condition" for imaging of mt-Keima. This is not standard practice and explains the discrepancy. Control fibroblasts have incredibly high levels of baseline mitophagy, which is unusual. I predict that with the author's experimental setup, i.e. transient transfection using lipids and differential laser settings, they would see the same result for any organelle labelled with this reporter.

There also appear to be aberrations within the images (a clear case of z-drift or misalignment in the photo presented in Figure 3Fi). From the extensive use of such, they should not look like this - especially with Airyscan or other high-quality imaging setups.

10. The authors have an outdated understanding of the field and assert "mitophagy is absolutely required" – and nothing else. Macroautophagy and mitophagy are very different processes. An increase in macroautophagy means elevated non-selective bulk turnover. Thus, using a mitophagy reporter – their increase in the signal can be attributed to this. However, it does not mean that the selective autophagy of mitochondria is actively occurring (i.e. the definition of mitophagy). There is no data on the receptor mediating the turnover of mitochondria or how its induction would differ from macroautophagy. As it stands, these terms are collectively bundled together and used interchangeably, and it isn't very pleasant to read. There is a wealth of literature on this that should be consulted.

11. The authors mention that mitophagy has been studied in the context of combined antimycin/oligomycin treatment. Yet no further information is provided. This would have serious implications for mechanistic understanding if mitophagy was in fact affected.

12. How does chronic AKT activation lead to mitochondrial elimination, and how is this different from macroautophagy?

There are also several confusing statements concerning autophagy in the paper: "suppresses macromolecular catabolism by autophagy" – this sentence is confusing. Autophagy is, by nature, a self-catabolic process. The induction of macroautophagy is the induction of non-selective catabolism.

In terms of the rebuttal letter, the Senior author writes that the "Inhibition by chloroquine may not be a fashionable way to inhibit mitophagy but its mode of action is not controversial". This reviewer would like to highlight that this assumption is, in fact, very incorrect. The mechanism of CQ and HCQ was actually a contentious issue in the autophagy field until it was finally resolved that it inhibits fusion without any effect on lysosomal pH (PMID: 29940786 - almost 700 citations since 2018).

This reviewer thinks that the manuscript is exciting but is undermined the problematic focus on autophagy. Without this data, the story would flow and make sense, and this reviewer would support its publication. In its current form, the autophagy data has been acquired and interpreted strangely. This does not mean these processes are not involved (although there are no differences in the key assays). The discovery of chronic AKT activation in mitochondrial disease is novel and would merit publication. However, far more compelling and rigorous data would be required to justify any credible invocation of macroautophagy or mitophagy here. The current data do not support this as a mechanism.

We have accepted this view and have adapted the paper accordingly. We hope that the result will prove acceptable.

Reviewer #2 (Remarks to the Author):

I think the revised manuscript has improved. I do not have more questions.

Reviewer #3 (Remarks to the Author)

The authors have addressed majority of my concerns; however, I would still argue that the inhibition PI3K-Akt-mTORC1 axis results in a partial rescue of the cellular bioenergetic function and the authors should report the rescue as "partial" throughout the manuscript, including the abstract.

We feel that to write 'partial rescue' at every mention is cumbersome. Further, the specific interpretation and description depends on what parameter you choose as a readout. For example, rescue of mitochondrial membrane potential was complete; rescue of respiratory rate was partial to baseline but complete to maximal respiratory capacity. Most, but not all, supercomplex assembly was completely rescued. We have therefore modified the text so that we vary the language as appropriate to the data set – we variably refer to 'partial rescue', 'improve', or the less contentious 'alleviate' or 'ameliorate' where rescue is partial, and to 'rescue' where it is complete.

The manuscript is well written and easier to read in this form after leaving some data out, yet some small issues need to be addressed before acceptance:

We are very pleased that the reviewer finds the work easier to read – the text has been edited extensively.

The authors show a high mitophagic flux in control fibroblasts in Fig. 3G/H? Why do so many control cells undergo mitophagy? Were these cells somehow treated to induce mitophagy? What happens if mitophagy is inhibited by CQ or Bafilomycin?

The cells were not treated to induce mitophagy – this is simply the resting rate in this cell type. Investigation of inhibition of mitophagy in control cells is a different study, but this is something that has been done in the past and we know that it results in dysfunctional mitochondria as do pathophysiological defects in mitophagy (e.g. see Osellame et al, PMID: 23707074). As we have now excluded these data on the insistence of reviewer 1, this no longer seems to be an issue.

The treatment of patient and cybrid cells with inhibitors of PI3K-Akt-mTORC1 axis, clearly improved the OXPHOS capacity, cells growth and supercomplex assembly, however as no treatment of control cells was done in parallel, it is impossible to assess if the partial rescue was achieved due to an overall mitochondrial biogenesis? The increase of Complex II suggests that the used drug might have such an effect. Please, comment.

It seems logical that there must be some biogenesis as there is increased mitophagy but mtDNA copy number remains unchanged. However, we have measured TFAM and found no change in expression level. We did treat the control cells in response to the previous review round and saw no effect on the variables we are studying.

Fig. 5B is very difficult to understand. What shows the pretreatment and what shows the treatment? Is it possible to change the graphs and group 0-pretreatment together and show within that group the effect of the treatment, then 4-weeks pretreatment, etc.? Also, how long was the treatment/measurement? Is it possible that all the high mutant load cells die within the first 24hours, and the expansion of lower mutant cells is propagated further? Would it be possible to pick a clone with a high load (at least in the case of the cybrids) and perform this experiment on a clone, to really prove this point? SytoGreen stain labels both live and dead eukaryotic cells, how was it used to measure dead cells?

We have tried to clarify the figure (Figs. 6a-b; lines 284-287 and 548-557). We thank the reviewer for the constructive suggestion. However, although separating and grouping data based on the periods of pretreatment may makes the graph simple, it seems to lose the information about the changes in cell growth rate/death ratio over the whole period of drug treatment, which is an important message that we have tried to convey here.

We took images every 2 h for a 3-4 day period for the measurements and the more details can be found in the Methods (lines 875-886).

If it this is true that the high mutant load cells die within the first 24h while cells with a lower mutant load expand, we would expect to observe a faster drop of mutant load in the mutant cells. However, the decrease in mutant load developed only after a few weeks of treatment. Also, our analysis for the cell death dynamics in A549 cybrid cells (since we took images every 2 h) showed a very similar pattern between the untreated and drug-treated groups in the first two days. After that, the death ratios in the drug-treated groups constantly dropped, while in the untreated group the death ratios gradually increased (please see below). Therefore, it is highly unlikely that the drug treatment kills the cells with high mutant loads first and allows the expansion of lower mutant cells afterwards.

Cell Death Dynamics

We did address this question in the previous round of reviews and reiterate that there is no evidence for increased cell death at all with the drug treatments, and we have seen a heteroplasmy shift at the single cell level, so we believe that we have excluded this explanation. Even picking a clone may not answer the question as when that expands it is likely to generate a population with mixed levels of heteroplasmy.

There was an embarrassing error here – as we used Sytox Green, which labels dead cells, not Syto Green and we thank the reviewer for spotting this mistake.

Fig. 6A - The autophagic flux is always increased by CQ treatment (e.g. Fig. 3E), as it prevents conversion to autolysosome. The LC3BII/LC3BI ratio between untreated and LY or RP treated cells should be compared to assess the autophagic flux, and this is not significantly increased based on this figure.

We thank the reviewer for spotting this error. However, as we have now excluded these data on the insistence of reviewer 1, this no longer seems to be an issue. Just for information - For Fig. 6A (and S7B) the LC3BII/LC3BI ratios of cells treated with CQ were not normalised the untreated controls but to the CQ-treated control, which is what we did for the first submission. During pervious revision, we re-analysed some of the immunoblotting data, normalising them to the untreated controls but forgot to do so for the data in Figs. 6A and S7B.

Fig. 6B, C – treatment of patient cells with RP and LY restores authophagic and mitophagic flux within 24 hours, however the decrease in mutant load does not happen until 4 weeks of treatment. If mitophagy starts happening at this point, why does the mutant load persist? Is mitophagy a very inefficient process?

This seems to us an interesting conceptual question but hard to answer experimentally – all we can do is to report what we have found. Again, as the mitophagy data have been removed, this is no longer an issue for the manuscript.

Minor comments:

Fig 3C-D - only one patient biopsy was used, however the figure legend and in D) figure description gives an impression that multiple patient biopsies were used: it should be corrected. What % of fibers was p-Akt/p-S6 positive in the patient biopsy?

We have used biopsies from two patients. All patient derived fibres were p-Akt positive but some were more intensely stained than others. There was a Gaussian distribution of intensities across individual fibres and the values shown in the histogram are the mean values.

Fig. 3E, S4C, 6A, S7B/E/H - two different sets of samples are shown, however only one loading control set. Is the loading control of cells treated or untreated with a drug? Please, provide a loading control for both sample sets. If only LC3BII/LC3BI ratio is calculated, the loading controls are not necessary, and could be removed?

We thank the reviewer for the constructive suggestion. However, as we have now excluded these data on the insistence of reviewer 1, this no longer seems to be an issue.

Line 236 – “showed an increased conversion to LC3BI to the lipidated LC3BII form”, should be ...”conversion of LC3BI to the lipidated LC3BII form”

‘Lipidated LC3BII’ seems correct. However, as we have now excluded these data on the insistence of reviewer 1, this no longer seems to be an issue.

Fig. 5C-D – in the main text the authors talk about a treatment of 6 and 12 weeks, however in the figure legend it is 4 and 8 weeks. Which is correct?

This has been corrected (lines 290-293). We thank the reviewer for spotting this mistake.

Discussion – “In contrast, in the present study, while we did not find changes in these two genes in the patient fibroblasts, we did find upregulation of FGF16 (~33 fold), suggesting that changes in FGF pathways in response to mitochondrial dysfunction might be tissue specific.” Was FGF16 increased in the cybrids? Please, correct the sentence.

We have revised the sentence (lines 401-403).

“In contrast, in the present study, while we did not find changes in these two genes in our RNA-seq data, we did find upregulation of FGF16 (~33 fold) in the patient fibroblasts, suggesting that changes in FGF pathways in response to mitochondrial dysfunction might be tissue specific.”

REVIEWERS' COMMENTS

Reviewer #1 (Remarks to the Author):

In general, the revised manuscript is much clearer, and with the more streamlined focus, the concerns of this reviewer have been satisfied. Congratulations to the authors on the exciting study.

Minor comments to the authors for technical clarification:

The authors cite antibody #2532 from Cell Signaling Technology as their phospho-specific AMPK antibody. CST #2532 does not detect phospho-AMPK but total levels of the alpha-1 subunit. This may be a simple typo, but can the authors please clarify which antibody they used to detect the modified form of AMPK? Perhaps the authors mean antibody CST #2531? This should be amended in the final version.

Related to this, the p-AMPK blot in S5 looks to have been reprobed. This approach is acceptable if the membrane was probed first with the phospho-specific antibody and then followed by the total. This may be obvious, but the abundance of the total protein can be far greater than the percentage that is phosphorylated. Thus, the incorrect detection sequence can lead to an artefact when detecting modified kinases, as the total protein is always tough to remove entirely by stripping. This reviewer highlights this particular point because the 15 kDa bands appear identical in terms of pixel features and intensity, yet the molecular weight markers are somehow different between these membranes/images? The authors should clarify whether these are the same or different membranes. Perhaps the authors have run a distinct substrate of AMPK and could include this, e.g. ACC? This is not required but would certainly add more weight to the claim of AMPK activation. The detection sequence should be clarified and stated in the methods to offset any concerns.

The antibody details for total and phospho-mTOR are not stated, and the detected band looks much lower in weight than would be typically expected. In the figure, the migration of mTOR appears around 200, whereas it is generally seen closer to 300 (289 kDa). Also, which phospho-site does the antibody detect? This information makes a difference to the accuracy of the paper and should be updated in the final version.

The blot for S6 in Fig S5a has an unusual feature – the middle molecular weight marker is somehow 'split' with a clean vertical line? Are these two different membranes? Th

If multiple antibodies were incubated with the membrane at the same time (multiplex detection, which is fine) – this should be stated in the methods. Doing so would explain the common features between each picture.

“For immunoblotting, cells were seeded 1-day prior experiments (60 mm plates for the A549 cybrid 879 cells and 10 cm plates for fibroblasts). To assess autophagic flux, cells were replenished with regular media for 1 h to prevent starvation-induced autophagy. After 1 h, treatment conditions resumed either with or without 50 μ M chloroquine for 6 h”.

Perhaps irrelevant as the authors have stated they are not reporting autophagy data, but this is not a conventional way to assess flux, mainly as there are no striking changes in autophagy factors or proteins in the data presented. If autophagy were the mechanism being investigated, the authors should be interested in inducing starvation-induced macroautophagy as differences may manifest more clearly here. Then \pm CQ would be essential to reveal the magnitude of any effect on induced flux. This text should be corrected/deleted to be consistent with the revised focus. The decision to proceed without autophagy/mitophagy is the right decision by the Senior author, particularly given the concerns on the previous data and as inspection of the present full-length blots confirms no differences in total mitochondrial content (e.g. levels of PDH) or changes of autophagy-related proteins (e.g. lipidated LC3 upon CQ treatment). As previously mentioned, steady-state LC3 levels are well known to differ between individual mammalian subjects, but this does not mean differential autophagy. Again, the manuscript is much improved without these data.

Minor text comments:

The authors should recheck the numbering of the supplemental figures and that they match the text?

“Fig. 4. The PI3K-Akt-mTORC1 axis is upregulated in the m.3243A>G mutant cells, perturbing 509 autophagy and mitophagy.”

“Supplementary Fig. 5. Other supporting results from patient fibroblasts and A549 cybrid cells for 650 assessing autophagy and mitophagy.”

As there is no mitophagy data in the paper (also according to the rebuttal letter), the authors should correct these titles.

Reviewer #3 (Remarks to the Author):

The authors have revised the manuscript and satisfied all my concerns.

REVIEWERS' COMMENTS

Reviewer #1 (Remarks to the Author):

In general, the revised manuscript is much clearer, and with the more streamlined focus, the concerns of this reviewer have been satisfied. Congratulations to the authors on the exciting study.

We thank the reviewer.

Minor comments to the authors for technical clarification:

The authors cite antibody #2532 from Cell Signaling Technology as their phospho-specific AMPK antibody. CST #2532 does not detect phospho-AMPK but total levels of the alpha-1 subunit. This may be a simple typo, but can the authors please clarify which antibody they used to detect the modified form of AMPK? Perhaps the authors mean antibody CST #2531? This should be amended in the final version.

We thank the reviewer for spotting this error. It has now been corrected as “anti-p-AMPK α (1:1000, Cell Signaling Technology #2535), anti-AMPK α (1:3000, Cell Signaling Technology #2532)” (line 599-600), which matched the details of materials in Supplementary Table 4.

Related to this, the p-AMPK blot in S5 looks to have been reprobed. This approach is acceptable if the membrane was probed first with the phospho-specific antibody and then followed by the total. This may be obvious, but the abundance of the total protein can be far greater than the percentage that is phosphorylated. Thus, the incorrect detection sequence can lead to an artefact when detecting modified kinases, as the total protein is always tough to remove entirely by stripping. This reviewer highlights this particular point because the 15 kDa bands appear identical in terms of pixel features and intensity, yet the molecular weight markers are somehow different between these membranes/images? The authors should clarify whether these are the same or different membranes. Perhaps the authors have run a distinct substrate of AMPK and could include this, e.g. ACC? This is not required but would certainly add more weight to the claim of AMPK activation. The detection sequence should be clarified and stated in the methods to offset any concerns.

This was done as the reviewer has surmised and we have now described the detection sequence in the methods (line 590-592).

The antibody details for total and phospho-mTOR are not stated, and the detected band looks much lower in weight than would be typically expected. In the figure, the migration of mTOR appears around 200, whereas it is generally seen closer to 300 (289 kDa). Also, which phospho-site does the antibody detect? This information makes a difference to the accuracy of the paper and should be updated in the final version.

We have now added the details of antibodies for total and phospho-mTOR (S2448) in Methods (line 596-597) and Supplementary Table 4. Phosphorylation of mTOR at S2448 is via the PI3K/Akt signalling pathway, which supports our hypothesis.

Since we initially wanted to detect multiple proteins with a huge range of molecular weights at the same time, we used 4-12% NuPAGE Bis-Tris polyacrylamide gels, in which the resolution for separating proteins is limiting. Also, the bands are the only bands that were about/above 200 kDa, we, therefore, believed they were p-mTOR/mTOR.

The blot for S6 in Fig S5a has an unusual feature – the middle molecular weight marker is somehow 'split' with a clean vertical line? Are these two different membranes?

We thank the reviewer for spotting this. They are one membrane which was cut into two for different protein targets at the protein transfer stage. We have added a note in the figure to avoid misunderstandings.

If multiple antibodies were incubated with the membrane at the same time (multiplex detection, which is fine) – this should be stated in the methods. Doing so would explain the common features between each picture.

We have now stated that multiple antibodies were incubated with membranes at the same time and referred to Supplementary data of immunoblotting raw images for details (line 586-587).

“For immunoblotting, cells were seeded 1-day prior experiments (60 mm plates for the A549 cybrid 879 cells and 10 cm plates for fibroblasts). To assess autophagic flux, cells were replenished with regular media for 1 h to prevent starvation-induced autophagy. After 1 h, treatment conditions resumed either with or without 50 μ M chloroquine for 6 h”.

Perhaps irrelevant as the authors have stated they are not reporting autophagy data, but this is not a conventional way to assess flux, mainly as there are no striking changes in autophagy factors or proteins in the data presented. If autophagy were the mechanism being investigated, the authors should be interested in inducing starvation-induced macroautophagy as differences may manifest more clearly here. Then \pm CQ would be essential to reveal the magnitude of any effect on induced flux. This text should be corrected/deleted to be consistent with the revised focus. The decision to proceed without autophagy/mitophagy is the right decision by the Senior author, particularly given the concerns on the previous data and as inspection of the present full-length blots confirms no differences in total mitochondrial content (e.g. levels of PDH) or changes of autophagy-related proteins (e.g. lipidated LC3 upon CQ treatment). As previously mentioned, steady-state LC3 levels are well known to differ between individual mammalian subjects, but this does not mean differential autophagy. Again, the manuscript is much improved without these data.

We thank the reviewer for noticing this. It seems clear that we need to review our data set relating to autophagy/mitophagy carefully. This part has now been removed (line 740).

Minor text comments:

The authors should recheck the numbering of the supplemental figures and that they match the text?

“Fig. 4. The PI3K-Akt-mTORC1 axis is upregulated in the m.3243A>G mutant cells, perturbing 509 autophagy and mitophagy.”

We thank the reviewer for noticing this. It has now been corrected as “The PI3K-Akt-mTORC1 axis is upregulated in the m.3243A>G mutant cells.” (line 962).

“Supplementary Fig. 5. Other supporting results from patient fibroblasts and A549 cybrid cells for 650 assessing autophagy and mitophagy.”

This has now been corrected as “Other supporting results from patient fibroblasts and A549 cybrid cells for the activity of the PI3K-Akt-mTORC1 axis”.

As there is no mitophagy data in the paper (also according to the rebuttal letter), the authors should correct these titles.

Reviewer #3 (Remarks to the Author):

The authors have revised the manuscript and satisfied all my concerns.